# Parallel functional testing identifies enhancers active in early postnatal mouse brain

Jason T Lambert[1,2†], Linda Su-Feher[1,2†], Karol Cichewicz[1,2], Tracy L Warren[1,2], Iva Zdilar[1,2], Yurong Wang[1,2], Kenneth J Lim[1,2], Jessica L Haigh[1,2], Sarah J Morse[1,2], Cesar P Canales[1,2], Tyler W Stradleigh[1,2], Erika Castillo Palacios[1,2], Viktoria Haghani[1,2], Spencer D Moss[1,2], Hannah Parolini[1,2], Diana Quintero[1,2], Diwash Shrestha[1,2], Daniel Vogt[3], Leah C Byrne[4,5], Alex S Nord[1,2]*

[1]Department of Psychiatry and Behavioral Sciences, University of California, Davis, Davis, United States; [2]Department of Neurobiology, Physiology and Behavior, University of California, Davis, Davis, United States; [3]Department of Pediatrics and Human Development, Grand Rapids Research Center, Michigan State University, Grand Rapids, United States; [4]Helen Wills Neuroscience Institute, University of California, Berkeley, Berkeley, United States; [5]Departments of Ophthalmology and Neurobiology, University of Pittsburgh, Pittsburgh, United States

*For correspondence: asnord@ucdavis.edu

†These authors contributed equally to this work

Competing interest: The authors declare that no competing interests exist.

**ABSTRACT** Enhancers are *cis*-regulatory elements that play critical regulatory roles in modulating developmental transcription programs and driving cell-type-specific and context-dependent gene expression in the brain. The development of massively parallel reporter assays (MPRAs) has enabled high-throughput functional screening of candidate DNA sequences for enhancer activity. Tissue-specific screening of in vivo enhancer function at scale has the potential to greatly expand our understanding of the role of non-coding sequences in development, evolution, and disease. Here, we adapted a self-transcribing regulatory element MPRA strategy for delivery to early postnatal mouse brain via recombinant adeno-associated virus (rAAV). We identified and validated putative enhancers capable of driving reporter gene expression in mouse forebrain, including regulatory elements within an intronic *CACNA1C* linkage disequilibrium block associated with risk in neuropsychiatric disorder genetic studies. Paired screening and single enhancer in vivo functional testing, as we show here, represents a powerful approach towards characterizing regulatory activity of enhancers and understanding how enhancer sequences organize gene expression in the brain.

## Introduction

*Cis*-regulatory elements such as enhancers are critical drivers of spatiotemporal gene expression within the developing and mature brain (*Nord, 2013*). Enhancers integrate the combinatorial functions of transcription factors (*Fuxman Bass et al., 2015*) and chromatin organizers (*Calo and Wysocka, 2013*) to drive cell and regional-specific expression of genes across brain development. DNA sequence variation within enhancers has been associated with both evolution (*McLean et al., 2011*) and the genetic etiology of neurological disorders (*Dunham, 2012*; *Perenthaler et al., 2019*; *D'haene and Vergult, 2021*). The first identified enhancers were defined by functional capacity to amplify transcriptional activity in reporter plasmids (*Moreau et al., 1981*; *Banerji et al., 1981*). Putative enhancers have been predicted via DNA conservation using comparative genomics and, more recently, by epigenetic signatures such as open chromatin from DNaseI hypersensitive site sequencing (DNase-seq) or assaying for transposase-accessible chromatin using sequencing (ATAC-seq) (*Dunham, 2012*), histone

tail modifications from chromatin immunoprecipitation sequencing (ChIP-seq) (*Nord, 2013*; *Visel, 2013*), and 3D chromatin organization (*Schoenfelder and Fraser, 2019*); however, these approaches are proxy measurements that do not directly evaluate whether a DNA sequence acts as a functional enhancer (*Benton et al., 2019*).

Enhancer reporter assays assess the ability of candidate DNA sequences to drive expression of a reporter gene, and have been used as the primary means of functionally testing activity of predicted enhancers (*Kvon, 2015*). Transgenic mouse enhancer assays have been applied to characterize the regulatory activity of candidate DNA sequences in the mouse, in both the developing and mature brain (*Nord, 2013*; *Visel, 2013*; *Silberberg, 2016*), and recently viral vectors have been used to deliver enhancer reporter constructs to mouse brain as well (*Graybuck et al., 2021*; *Mich, 2021*). These assays offer exceptional information on tissue-specific enhancer activity but require one-by-one testing of individual candidates. The advancement of massively parallel reporter assays (MPRAs) has led to the ability to functionally screen a multitude of DNA sequences for enhancer activity in parallel within a single experiment using sequencing-based quantification (*Inoue and Ahituv, 2015*). To date, there are a few promising demonstrations of MPRA applied in vivo in brain (*Shen et al., 2016*; *Shen, 2019*), although screening approaches to characterize enhancers remain highly novel, particularly in developing brain. Functional assessment of enhancer activity and the use of enhancers to express reporters and genes in mouse brain cells has emerged as an area of major interest (*Dimidschstein et al., 2016*; *Hrvatin, 2019*; *Nord and West, 2020*; *Vogt, 2014*). In vivo screening approaches represent an ideal method to perform broad testing at scale and maintain tissue-type and developmental context.

Here, we adapted an MPRA functional enhancer screen, self-transcribing active regulatory region sequencing (STARR-seq) (*Arnold et al., 2013*), for application to early postnatal mouse brain via recombinant adeno-associated virus (rAAV) delivery. We screened a library of candidate human DNA sequences spanning putative brain-specific regulatory sequences and regions associated with genome-wide association studies of epilepsy and schizophrenia. We identified sequences capable of enhancing reporter expression in early postnatal mouse cortex, and validated positive and negative MPRA expression predictions, including within the neuropsychiatric disorder-associated third intron of *CACNA1C* (*Moon et al., 2018*; *Ferreira et al., 2008*; *Sklar, 2008*). Our results demonstrate the utility of parallel functional testing to interrogate the regulatory activity of enhancers in early postnatal mouse brain and highlight opportunities for functional screening in studies of normal brain development and function as well as the etiology of neurodevelopmental and neuropsychiatric disorders.

## Results
### Developing and validating AAV-MPRA strategy
We used a modified STARR-seq (*Arnold et al., 2013*) MPRA orientation, in which the candidates of interest were cloned in the 3' untranslated region (3'-UTR) of a reporter gene (here EGFP) driven by the *Hsp1a* minimal promoter (HspMinP; formerly referred to as Hsp68 minP *Bevilacqua et al., 1995*). This reporter construct was flanked by inverted terminal repeat regions (ITRs) necessary for packaging self-complementary adeno-associated virus (scAAV), which has the advantage of faster transduction rates compared to conventional AAV (*McCarty, 2008*; *Figure 1A*, *Figure 1—figure supplement 1*). Enhancers are proposed to function independent of orientation and the position relative to the transcriptional start site (*Serfling et al., 1985*), as has been demonstrated in in vitro applications of STARR-seq (*Arnold et al., 2013*; *Klein et al., 2020*). However, we first sought to verify that an enhancer sequence in the STARR-seq orientation not only increased transcription generally, but also did so in a cell-type defined restricted manner in vivo when paired with HspMinP and delivered via scAAV. To validate this, we cloned a GABAergic-biased mouse *Dlx5/6* enhancer (*Dlx*) (*Dimidschstein et al., 2016*; *Lee et al., 2014*) into the scAAV reporter vector. scAAV9 carrying HspMinP-EGFP-*Dlx* was mixed with AAV9 carrying the constitutively active CAG-mRuby3 reporter and this mixture was delivered to postnatal day (P)0 mouse forebrain via brain intraventricular injection and collected at P7. The expression of mRuby3 was used as a positive control and to locate the injection site for analysis. Red fluorescent cells, marking the focus of viral exposure, were found distributed in the cortex and hippocampus (*Figure 1B*, *Figure 1—figure supplement 2A, B*). We analyzed EGFP-expressing

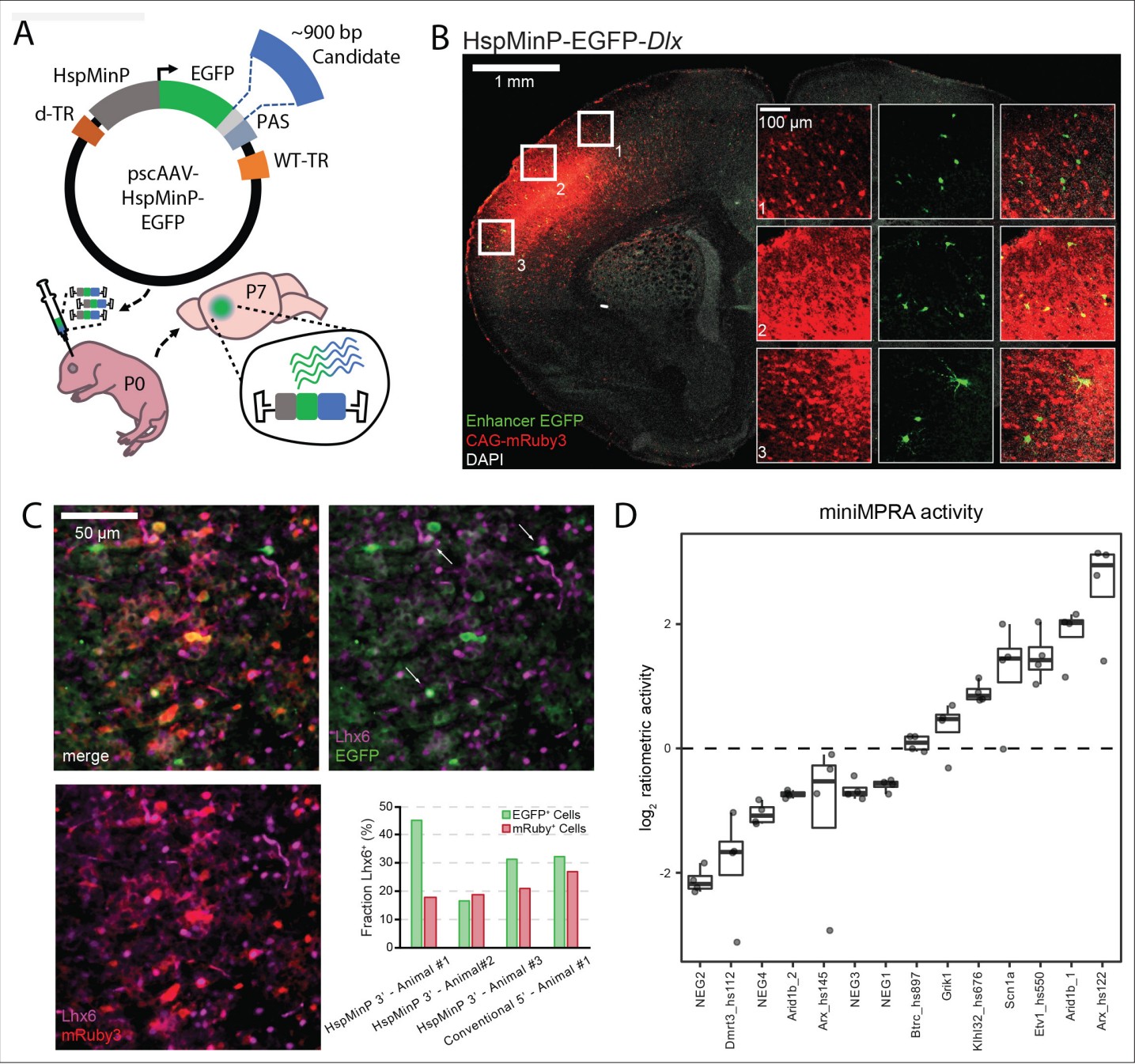

**Figure 1.** Designing and validating 3'-UTR enhancer reporter AAV assay. (**A**) Schematic of in vivo parallelized functional enhancer reporter assay. The test library was generated using the previral vector pscAAV-HspMinP-EGFP, which contained a multiple cloning site (light grey) between the EGFP reporter and polyadenylation site (PAS). Purified PCR products for test amplicons were cloned into the vector using Gibson assembly. The previral library was packaged into AAV9(2YF), and the viral library delivered to the brain via injection at P0. Brains were collected at P7. (**B**) Representative image of a coronal section of a P7 mouse brain injected at P0 with a virus mixture consisting of an AAV containing the STARR-seq vector carrying the inhibitory interneuron enhancer *Dlx* (scAAV9-HspMinP-EGFP-*Dlx*) and an injection control AAV containing an expression vector for mRuby3 under the control of CAG, a general mammalian promoter. EGFP expression was visualized via IHC using an anti-GFP antibody, while mRuby3 expression was visualized using native fluorescence. Insets show close up of boxed regions showing morphology of EGFP-expressing cells in the cortex. (**C**) Sections from P7 mouse cortex transduced with *Dlx*-driven STARR-seq reporter vector and mRuby3 injection control at P0, counterstained with an antibody for Lhx6, a transcription factor active in deep cortical layer interneurons. EGFP-expressing cells with Lhx6+ Nuclei are indicated with arrows. Inset graph shows fraction of EGFP- or mRuby3-expressing cells co-labeled with Lhx6 in three replicate animals injected with scAAV9-HspMinP-EGFP-*Dlx* (Animal 1, n = 20 EGFP+ cells, 218 mRuby3+ cells; Animal 2, n = 18 EGFP+ cells, 435 mRuby3+ cells; Animal 3, n = 32 EGFP+ cells, 311 mRuby3+ cells) or one animal injected with AAV9-*Dlx*-β GlobinMinP-EGFP (***Dimidschstein et al., 2016***; ***Lee et al., 2014***) (n = 31 EGFP+ cells, 63 mRuby3+ cells). (**D**) Ratiometric (log₂

*Figure 1 continued on next page*

*Figure 1 continued*

RNA/DNA) activity of miniMPRA mouse library in P7 mouse cortex after injection at P0. Boxplot of distribution and individual replicates (N = 4) shown for the 16 tested candidates. NEG indicates putative negative candidate; otherwise, name indicates nearby gene and if applicable, embryonic enhancer ID, for positive candidates.

The online version of this article includes the following figure supplement(s) for figure 1:

**Figure supplement 1.** Vector map of pscAAV-HspMinP-EGFP.

**Figure supplement 2.** *Dlx*-driven EGFP expression in excitatory vs inhibitory cells based on cellular morphology in the cortex and hippocampus.

**Figure supplement 3.** Additional biological replicates of Lhx6 co-staining of *Dlx*-driven EGFP⁺ cells.

**Figure supplement 4.** Interneuronal expression of *Dlx*-driven EGFP expression is consistent despite variable transductions.

cells in these regions. The brains were fixed, cryosectioned, and stained with an antibody for GFP to enhance detection of *Dlx* activity (see Materials and methods).

We evaluated cell-type specificity two ways: by assessing morphology and by counterstaining with an antibody for Lhx6, a cardinal transcription factor expressed in both developing and mature neurons derived from the medial ganglionic eminence (**Sandberg, 2016**; **Alifragis et al., 2004**). EGFP⁺ cells in these P7 brains had small cell somas and neurites, typical of interneurons compared to pyramidal cells and glia (**Figure 1B**, **Figure 1—figure supplement 2A, B**); many were Lhx6⁺, but this was quite variable across animals (**Figure 1C**, **Figure 1—figure supplement 3**). We noted that the EGFP⁺ cells that were negative for Lhx6 staining still primarily appeared to be interneurons based on morphology. Since *Dlx* should be active in multiple GABAergic interneuron types in the neocortex (**Dimidschstein et al., 2016**), we systematically analyzed the morphology of EGFP⁺ cells (**Figure 1—figure supplement 2**). The principal excitatory neurons in the cortex and hippocampus have a typical pyramidal shape with a tear-drop cell body and a long, primary apical dendrite, whereas inhibitory interneurons are more variable in their appearance. Furthermore, excitatory neurons in the hippocampus are restricted to specific pyramidal and granule layers, and spatial location could be used as a second classification criterion there to lend extra confidence to the morphological assessment. Based on these morphological and spatial criteria, the vast majority of EGFP-expressing cells resemble interneurons (92.7 %, n = 314 cells in five animals, **Figure 1—figure supplement 2A, C**). 3.2 % of EGFP⁺ cells exhibited pyramidal morphology. We compared these results with expression driven from the same *Dlx* sequence in a conventional orientation upstream of a minimal β-Globin promotor, a construct that has previously been validated to exhibit interneuron-biased expression (**Dimidschstein et al., 2016**). The specificity for interneurons in our hands was similar in both constructs (**Figure 1—figure supplement 2B, C**). These data, taken together, indicate that enhancer orientation and choice of minimal promoter did not greatly affect cell-type-specific activity of the enhancer reporter construct.

We next performed a small-scale parallel reporter assay (miniMPRA), testing 16 mouse sequences representing putative enhancer and negative control sequences (**Figure 1D**, **Supplementary file 1**). Long sequences of interest (~900 bp) were selected to maximize the chromosomal context of candidate enhancer elements within these regions. These sequences were predicted to have either enhancer activity or no activity in postnatal mouse brain based on previous transgenic mouse enhancer assays and on mouse brain H3K27ac ChIP-seq data (**Nord, 2013**; **Visel, 2013**). We batch-cloned these sequences into the 3'-UTR of our scAAV reporter construct, and we packaged this into a scAAV library using AAV9(2YF) (**Dalkara et al., 2012**). AAV9(2YF) is a tyrosine-mutated derivative of AAV9, which has increased transduction in the brain and similar tropism compared to standard AAV9. We delivered the scAAV library to P0 mouse prefrontal cortex by direct injection and collected the brains for DNA and RNA analysis at P7 (**Figure 1A**). In this proof-of-principle experiment, putative negative control sequences displayed low transcription levels based on the ratio of RNA-seq read counts to DNA-seq read counts. On the other hand, previously validated embryonic brain enhancers (**Visel, 2013**) and putative enhancers of interest near genes associated with neurodevelopmental disorders or epilepsy (**Turner, 2016**; **Nakayama et al., 2010**; **Gao et al., 2017**) were found to be active as enhancers in this assay at P7 (**Figure 1D**). Based on this proof-of-principle experiment, we moved forward with scaling up our scAAV STARR-seq strategy to testing hundreds of sequences in mouse brain.

## Cloning and AAV packaging of MPRA library

We generated an MPRA library targeting 408 candidate human DNA sequences for testing, each ~900 bp in size, to assess functional enhancer activity in vivo in early postnatal mouse brain. These sequences were categorized into four groups (*Figure 2A*), two groups without ascertainment based on potential regulatory activity, and two groups selected based on presence of epigenomic signatures commonly used to predict enhancer activity (*Supplementary file 2*). For the first group ('GWAS'), we selected genomic intervals containing single-nucleotide polymorphisms (SNPs) identified as lead SNPs from genome-wide association studies (GWAS) for epilepsy (*Abou-Khalil, 2018*) and schizophrenia (*Ripke, 2014*) as identified using the NHGRI-EBI GWAS Catalog (*Buniello et al., 2019*). For the second group ('LD'), we selected five lead SNPs from these same studies, and included genomic intervals that included all SNPs in linkage disequilibrium (LD, $r^2$ >0.8, *Auton et al., 2015*) within these non-coding regions associated with the *CACNA1C* and *SCN1A* loci. Because SNPs can be inherited together, GWAS lead SNPs are not necessarily causal and further evidence is necessary to prioritize linked variants. Thus, most SNP-containing amplicons in these first two groups were expected to be negative for enhancer activity. Indeed, less than 20 % of GWAS and LD group sequences had strong evidence of open chromatin in fetal brain or adult neurons (*Figure 2A*, lower panel). For the third group ('FBDHS'), we selected regions overlapping DNaseI hypersensitive sites in fetal human brain from ENCODE (*Dunham, 2012*), selecting candidates within copy number variant regions near autism risk genes (*Turner, 2016*; *Turner et al., 2017*). We expected this group to be enriched for active enhancers in our assay, although enhancers active in fetal brain may not continue to have activity at the P7 postnatal time point tested in our assay. Finally, we included human orthologs of the 16 sequences tested in our miniMPRA experiment ('PutEnh'), which included putative positive and negative sequences. Thus our final library included a balance of sequences that are expected to have no enhancer activity (59 %), as well as sequences with evidence for regulatory capacity in the brain (41 %).

We cloned all sequences of interest (referred to from here on as amplicons) from pooled human DNA via polymerase chain reaction (PCR) and Gibson assembly directly into the viral DNA plasmid backbone. Following batch cloning, we sequenced the pooled plasmid library, verifying presence of 345/408 (84.6 %) target amplicons. There was no obvious pattern among amplicons that did not make it into the batch cloned library, and we presume differences in PCR primer performance and general stochasticity were the primary drivers of batch cloning success. This cloned library of candidates was then packaged in an scAAV9(2YF) viral vector in the same manner as our miniMPRA.

## Application of in vivo MPRA and assessment of reproducibility of biological replicates

We delivered the viral library to the left hemisphere of four neonatal mouse brains at P0 via direct injection into the prefrontal cortex and collected forebrain tissue seven days after transduction at P7. We isolated viral genomic DNA, representing input or delivery control, as well as total RNA, and generated amplicon sequencing libraries for the left hemispheres for each replicate as well as for the previral batch-cloned plasmid library. RNA samples were treated with DNase, to prevent cross-contamination, and tagmentation libraries were prepared to capture the 3'-UTR variable region. Following sequencing, de-duplicated aligned reads to the human genome were used to generate amplicon summary counts (*Supplementary file 3*). We then filtered the dataset by removing amplicons with raw counts less than 200 and mean amplicon library proportions less than $2^{-15}$ in any DNA sample, leaving 308/345 (89 %) passing these quality control thresholds. We observed significant correlation between biological replicates (Pearson correlation, p < 0.001), although correlation was consistently higher for DNA replicates compared to RNA replicates ($r$ > 0.824 for DNA comparisons, and $r$ > 0.546 for RNA comparisons, *Figure 2—figure supplement 1A, B*). Correlation within DNA or RNA was higher than between library type, suggesting limited, if any, impact of cross-contamination. We also observed strong correlation between amplicon read counts in the pooled previral plasmid library ('Library') and genomic AAV DNA ('DNA') collected from each injected P7 brain ($r$ > 0.804, p < 0.001, *Figure 2—figure supplement 1A*). To test the effect of PCR bias during library preparation on amplicon counts, we generated one technical RNA replicate, Sample 4–35, using a higher cycle count (*Figure 2—figure supplement 1B*), and found similar amplicon representation and dropout patterns as its matched lower-cycle technical replicate (Pearson $r$ = 0.827, p < 0.001). Taken together, this demonstrates that viral packaging, neonatal delivery, P7 sample collection and processing, library

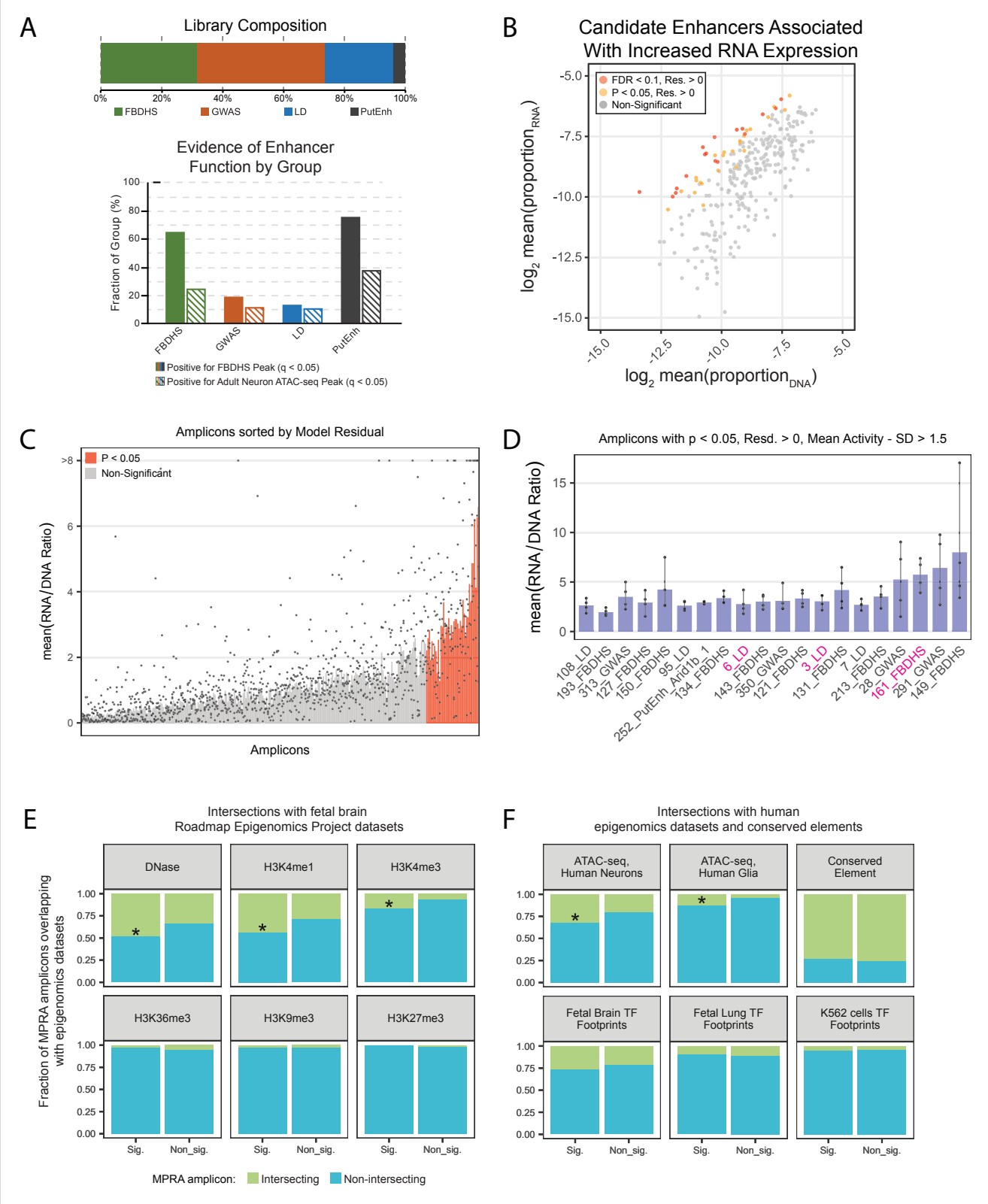

**Figure 2.** In vivo AAV MPRA yields amplicons capable of enhancing transcription, enriched for signatures associated with enhancers. (**A**) Graphical representation of library composition. Top panel shows the fraction of the total library made up by each group of amplicons. Bottom panel shows the fraction of amplicons in each group that were positive for the given epigenomic signature. (**B**) Mean RNA and DNA representation in the assay for candidates that passed inclusion criteria (N = 308). Amplicons with significantly (p < 0.05, FDR < 0.1) increased model residual value (Res.) in RNA

*Figure 2 continued on next page*

*Figure 2 continued*

compared to DNA are shown in orange and red. Normal p-values; empirical FDR q-values (See Materials and methods). (**C**) Bar plot representing mean activity based on RNA/DNA ratio in the test assay with individual replicates shown as dots. Amplicons are sorted by linear model residuals (p < 0.05 colored red). (**D**) The top 20 active amplicons with consistent activity across both linear and ratiometric models. Bars represent mean activity based on RNA/DNA ratio and individual replicates are shown as dots. Three amplicons were used for downstream validation in single-candidate deliveries (magenta). (**E**) Amplicon intersection with fetal brain epigenomic datasets including DNase Hypersensitive loci, H3K4me1, H3K4me3, H3K36me3, H3K9me3, and H3K27me3. Amplicons were divided into two groups based on the statistical significance of their activity in the MPRA. (**F**) Amplicon intersection with human neuron or glia ATAC-seq, vertebrate conserved elements, and digital transcription factor footprints in fetal brain, fetal lung, and K562 cells. Asterisks in E and F indicate significant enrichment for positive amplicons with annotation class (p < 0.05, permutation test).

The online version of this article includes the following figure supplement(s) for figure 2:

**Figure supplement 1.** Reproducibility of in vivo MPRA.

**Figure supplement 2.** Amplicon selection for linear model building.

**Figure supplement 3.** MPRA activity for all amplicons in the library.

**Figure supplement 4.** Allele-specific analysis of MPRA activity.

**Figure supplement 5.** In vivo AAV MPRA replicates findings of the miniMPRA.

generation, and sequencing did not substantially affect amplicon representation. Although correlation between amplicons generated from P7 cDNA ('RNA') was strong overall, particularly for amplicons with robust cDNA expression, each delivery replicate included a small subset of lower cDNA expression amplicons that showed replicate-specific cDNA dropout (*Figure 2—figure supplement 1B*). Amplicon representation in the previral library and P7 viral DNA was correlated with amplicon GC content (*Figure 2—figure supplement 2A-B*), indicating GC-based differences impact PCR and cloning efficiency. There was reduced strength in correlation between GC content and the MPRA RNA (*Figure 2—figure supplement 2C-D*) and no further GC bias arose between viral packaging, delivery, and recapture of the library, suggesting that GC bias originated from the batch cloning process. Input viral DNA and MPRA RNA amplicon counts were also generally correlated, indicating some basal MPRA transcription from the minimal promoter independent of candidate enhancer sequence activity, as reported in other MPRA studies (*Lee, 2020*; *Ashuach et al., 2019*).

## Identification of putative enhancer activity in P7 brain based on MPRA analysis

Having confirmed reproducibility of input viral DNA and MPRA RNA collection, we performed activity estimation. We used two approaches for activity estimates: first, regression residual-based estimate to correct for background or basal activity, and second, via simple comparison of RNA to DNA amplicon representation.

First, we used the middle 80 % of amplicons (N = 248) ranked by RNA/DNA ratio (*Figure 2—figure supplement 2E*) to build a linear model to account for background MPRA RNA based on amplicon representation and GC content (*Figure 2—figure supplement 2E-H*). For regression-based estimation of activity, amplicon representation across each replicate was combined to generate a summary activity value. To identify amplicons with activity above expected based on background (i.e. presumed enhancers), we applied the model to the full set of amplicons (N = 308) to generate residual values that represent observed RNA levels compared to expected. p-values for amplicon activity versus background were empirically defined using the distribution of standardized residuals (*Figure 2B*, *Figure 2—figure supplement 3*, *Supplementary file 4*). We identified 41 amplicons (13 %) with significantly increased RNA (p < 0.05), suggestive of positive activity, with 17 passing with a false discovery rate (FDR) < 0.1. Amplicon group was confirmed to be a significant predictor of mean amplicon activity (One-way ANOVA, p = 0.006), with the highest activity reported for the PutEnh and FBDHS groups.

Second, we also estimated activity based on RNA/DNA ratio (*Figure 2C*), a common metric used to report activity in published MPRAs (*Inoue and Ahituv, 2015*; *Klein et al., 2020*; *Ashuach et al., 2019*). We found that 71 amplicons were considered active using the criteria RNA/DNA ratio > 1.5 and RNA/DNA standard deviation less than its mean. Amplicons that exhibited RNA/DNA ratio > 1.5 but did not pass the regression model-based p-value potentially included sequences with weaker enhancer activity. 78 % (32/41) amplicons identified using model residuals were also active in the ratiometric

comparison. Among 41 positive amplicons with significant regression residual activity, 20 (49 %) had mean ratio – 1 s.d. >1.5 across individual replicates, representing the amplicons with the strongest and most consistent MPRA-defined activity (*Figure 2D*). For statistical testing of RNA/DNA ratios, we used a Wilcoxon rank sum, a non-parametric test comparing DNA vs RNA proportion values across delivery replicates. 43 amplicons passed $p < 0.05$ with confidence intervals greater than 0 (one-tailed for increased RNA), and all of them had Benjamini-Hochberg corrected FDR < 0.05. In comparison to the linear model approach, 26 amplicons were significant at $p < 0.05$ in the Wilcoxon rank sum test and in the linear model.

We compared the 41 model-based significant amplicons to the remainder of the library for intersection with fetal brain epigenomics datasets from the Roadmap Epigenomics Project (*Bernstein et al., 2010*). We found higher than expected intersection of significant amplicons with genomic loci characterized as DNase hypersensitive (DNase, $p = 0.0355$). We also observed that significant amplicons had increased enrichment for H3K4me1 ($p = 0.0274$), and H3K4me3 ($p = 0.0131$), epigenomic marks associated with transcriptional activity and enhancer function. We did not find enrichment for H3K36me3, a histone mark associated with gene bodies, or for H3K9me3 and H3K27me3, histone marks associated with heterochromatin (*Figure 2E*).

We also intersected amplicons with a human neuron and glia ATAC-seq dataset from the Brain Open Chromatin Atlas (BOCA) (*Fullard et al., 2018*), vertebrate evolutionary conserved elements (UCSC Genome Browser), and digital transcription factor (TF) footprints (*Vierstra et al., 2020*). Similar to *Figure 2E*, we found marginally significant increased overlap of significant amplicons with human neuron ATAC-seq peaks ($p = 0.0497$). We also found increased enrichment of significant amplicons in open chromatin in human glia ATAC-seq ($p = 0.0159$). There were no significant differences in enrichment among amplicon groups in TF footprints in fetal brain or lung, nor in K562 cell lines, nor for conserved elements (*Figure 2F*). We also examined each epigenetic mark as a co-variate of a general linear model, to see if it improved a regression model predicting cDNA levels considering all amplicons (N = 308). If cDNA levels in our MPRA are reflective of true enhancer activity, we would expect epigenomic signatures associated with enhancer elements to improve prediction of cDNA expression. Indeed, Fetal DNase hypersensitivity ($p = 0.0002$), H3K4me1 ($p = 0.0002$), adult neuron ATAC-seq ($p = 0.0073$), and fetal brain TF footprints ($p = 0.04$) were found to be significant, while signatures not relevant to enhancer activity in brain were not (*Supplementary files 5 and 6*). Fetal brain DNase hypersensitivity, H3K4me1, and human neuron ATAC-seq improved the regression model (reduced BIC), and fetal brain DNase hypersensitivity and H3K4me1 signatures had the strongest predictive power of amplicon activity (0.2 and 0.178 point-biserial correlations, respectively).

Based on the intersections with DNaseI hypersensitive and neuronal ATAC-seq signatures, there was a high representation of ascertained likely enhancers (20/41) among significantly active amplicons. These 20 active amplicons represented 15.9 % of the set of ascertained likely enhancers, while the remaining 21 active amplicons represented 11.5 % of predicted negative sequences. The difference in amplicons with MPRA activity (model-based p-value < 0.05) comparing ascertained positive and negative amplicon sets was not significant based on Fisher's exact test (odds ratio of 1.45, one-tailed p-value = 0.176). The representation of ascertained likely enhancers was higher among the most significant amplicons with FDR < 0.1 (12/17). These 12 amplicons represented 9.5 % of the ascertained likely enhancers, while the remaining five active amplicons represented 2.7 % of the predicted negative sequences, resulting in an odds ratio of 3.73. The enrichment of this highly significant set of MPRA amplicons for pre-ascertained likely enhancers was statistically significant (Fisher's exact test, one-tailed p-value = 0.011). Therefore, consistent with our expectations, enhancer ascertainment based on open chromatin patterns in the brain was associated with MPRA activity.

## Lack of strong effects and insufficient power for allele-specific activity analysis

Our MPRA library included a subset of amplicons harboring SNPs that were either lead SNPs from GWAS studies (121 amplicons), or SNPs in LD with lead SNPs within associated intervals at *CACNA1C* (21 amplicons) and *SCN1A* (62 amplicons). As amplicons were cloned from pooled human population DNA, we anticipated capturing representative alleles for these SNPs in our MPRA library. Indeed, 439/440 of these SNPs, representing 313/345 cloned amplicons, were represented in our library before viral packaging, but only 147 of them had minor allele frequency above 0.1, critical for allele

comparison. Among the eight amplicons with significant MPRA activity, none exhibited significant allelic differences (*Figure 2—figure supplement 4G*). We further used luciferase assays in two cell lines (HEK and SH-SY5Y) to compare haplotype activity for one MPRA active enhancer, finding no significant difference between the alleles of the *CACNA1C* SNP containing amplicon #3 (*Figure 2—figure supplement 4H*). We note that despite strong DNA reference allele frequency correlation in the viral library DNA, limited allele coverage, unbalanced allele frequencies, and reduced correlation of RNA allele frequencies resulted in lack of power to detect moderate to small allelic differences in our experiments (*Figure 2—figure supplement 4*, *Supplementary file 7*).

## Activity reproduces between full MPRA and miniMPRA experiments

Since the PutEnh group in our MPRA contained human orthologs of the miniMPRA sequences, this experiment enabled comparison of human and mouse sequences in an in vivo mouse brain context and offered further validation of reproducibility for our MPRA results. Eight amplicons were included in both experiments, and we observed general correlation between activity in the mouse miniMPRA and the human orthologs in the full MPRA library. These results further indicate reproducible activity and that there is conserved function between orthologous mouse and human sequences tested in the same postnatal mouse brain context (*Figure 2—figure supplement 5*).

## Confirmation of in vivo P7 cortex MPRA results for single candidate sequences

To validate enhancer activity in the mouse brain at P7, we next cloned individual amplicons for selected positive and negative hits from the MPRA into the same HspMinP-EGFP 3'-UTR oriented reporter and generated scAAV9 for each construct. Amplicon #161 (FBDHS group), an enhancer candidate that overlaps both a DNaseI hypersensitive site in fetal human brain and a copy number variant region near the autism- and epilepsy-associated gene *SCN2A*, and that displayed particularly strong activity in the screen (*Figure 3—figure supplement 1A*), showed consistent expression of EGFP in the mouse brain at P7 (*Figure 3A*, top row). EGFP expression driven by this amplicon was seen wherever the brain had been exposed to the virus (based on mRuby3 expression), with a trend toward greater density of EGFP$^+$ cells observed in the lower-middle layers of the cortex (lower half of Layer IV and Layer V). On the other hand, a predicted negative sequence that did not display enhancer activity in the screen, amplicon #264, did not show expression of EGFP (*Figure 3A*, bottom row, *Figure 3—figure supplement 1B*).

To further validate enhancer function in excitatory glutamatergic neurons, we counter-stained for Ctip2, a transcription factor involved in axonal development in excitatory cortical projection neurons in Layer V of the cortex (*Arlotta et al., 2008*; *Leyva-Díaz and López-Bendito, 2013*). Although expression of Ctip2 is not exclusive to excitatory neurons in adult mice (*Nikouei et al., 2016*), it is commonly used as an excitatory Layer V marker during embryonic and early postnatal development (*Leyva-Díaz and López-Bendito, 2013*; *Alcamo et al., 2008*; *Leone et al., 2015*; *Gompers et al., 2017*). We reasoned that if our in vivo MPRA reporter construct accurately reproduced cell-type specific enhancer activity, then we should see a difference in the Ctip2 overlap with EGFP driven by amplicon #161 compared to EGFP driven by *Dlx*. Indeed, after counterstaining brain sections transduced as described above with antibodies for Ctip2, we found that EGFP$^+$ cells driven by *Dlx* exhibited significantly lower frequencies of Ctip2$^+$ nuclei compared to either CAG-driven mRuby3$^+$ cells or EGFP$^+$ cells driven by amplicon #161 (*Figure 3B and C*). These results demonstrate that our MPRA could accurately reflect enhancer activity of particular candidate sequences in vivo.

## Dissection of regulatory elements within the third intron of *CACNA1C*

Our library included amplicons spanning across a psychiatric disorder-associated LD interval within the ~330 kb third intron of the gene *CACNA1C*, which encodes the α1 subunit of the L-type voltage-gated calcium channel Ca$_V$1.2. This region contains previously in vitro defined regulatory elements (*Eckart et al., 2016*; *Roussos, 2014*) harboring schizophrenia- or bipolar disorder-associated SNPs, predominantly *rs1006737* (*Ferreira et al., 2008*; *Sklar, 2008*), *rs2007044* (*Ripke, 2014*), *rs4765905* (*Hamshere et al., 2013*), and *rs4765913* (*Ripke, 2014*). Via MPRA, we assessed the activity of 17 amplicons within the *CACNA1C* intron covering SNPs in LD (r$^2$ >0.8). As a comparison set, we also

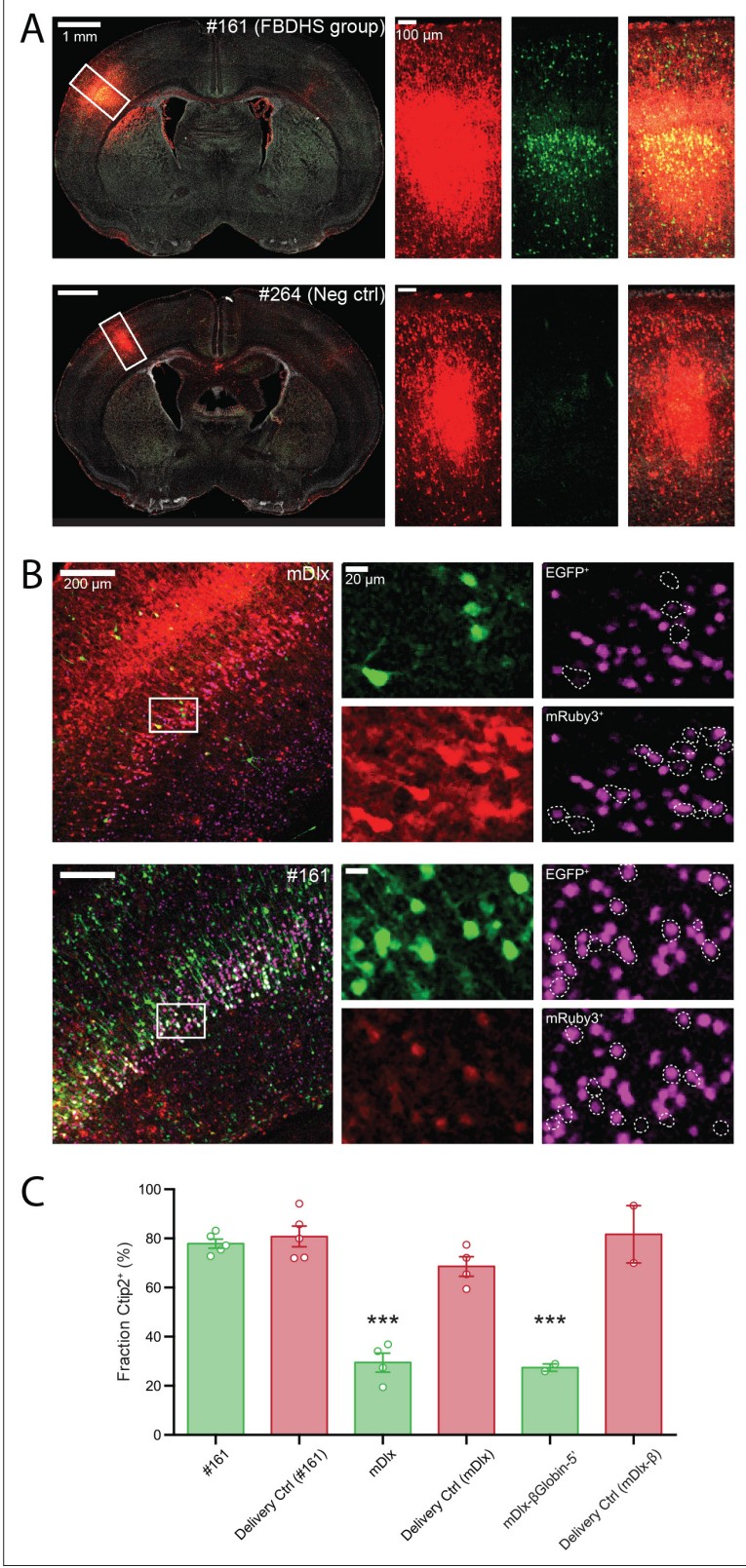

**Figure 3.** Functional validation of STARR-seq screen. (**A**) Validation of positive and negative hits from in vivo AAV MPRA screen. Representative images of coronal sections of AAV-transduced P7 brains stained with an anti-GFP antibody is shown (left panels). Closeup of the boxed regions are shown in the panels on the right (from left to right: Red channel, mRuby3 injection control; Green channel, EGFP expression driven by candidate amplicon;

*Figure 3 continued on next page*

*Figure 3 continued*

Merge with DAPI in gray). The brain shown in the top row was transduced with AAV9-CAG-mRuby3 (injection delivery control) and scAAV9-HspMinP-EGFP carrying in the 3'-UTR Amplicon #161, a highly active amplicon in the AAV MPRA. In the bottom row, a similar transduction is shown for Amplicon #264, a negative control with no predicted enhancer activity that did not display activity in the in vivo AAV MPRA. (**B**) Functional validation of enhancer activity in different cell types. Brains were transduced as in A with AAV9-CAG-mRuby3 and scAAV9-HspMinP-EGFP carrying either Amplicon #161 or *Dlx* in the 3'-UTR. Brains were collected at P7 and stained for GFP and Ctip2, a transcription factor necessary for axon development in excitatory projection neurons in Layer V during embryonic development. Representative staining of coronal sections is shown (left panels). Zoomed in views of boxed regions are shown in single-channel images in the panels on the right (Green, EGFP; Red, mRuby3; Magenta, Ctip2). Ctip2 channel images are shown with EGFP+ cells outlined (top) and mRuby3+ cells outlined (bottom). (**C**) Quantification of Ctip2 co-labeling shown in B with additional animals co-transduced with AAV9-CAG-mRuby3 and AAV9-*Dlx*-$\beta$ GlobinMinP-EGFP included for comparison. Individual GFP+ and mRuby3+ cells were counted and scored for whether each cell contained a Ctip2-positive nucleus. Cell counts were summed across images for the same brain. Data is presented as mean ± SEM for the fraction of fluorescent cells that are Ctip2+. Cells that expressed EGFP under the control of the inhibitory interneuron enhancer *Dlx* displayed a lower frequency of Ctip2+ nuclei compared to cells that drove EGFP under the control of amplicon #161 or drove mRuby3 under the control of the general mammalian promoter CAG (n = 5 animals co-injected with HspMinP-EGFP-#161 and CAG-mRuby3, four animals co-injected with HspMinP-EGFP-*Dlx* and CAG-mRuby3, and two animals co-injected with *Dlx*-$\beta$ GlobinMinP-EGFP and CAG-mRuby3).

The online version of this article includes the following figure supplement(s) for figure 3:

**Figure supplement 1.** MPRA results for validated single candidates in genomic context.

**Figure supplement 2.** Replicates for validation of positive and negative MPRA hits.

included five amplicons covering SNPs in LD (r² >0.8) associated with a non-neuronal SNP associated with hematocrit (*rs7312105*) (**van der Harst et al., 2012**).

Three amplicons within the *CACNA1C* psychiatric disorder LD interval drove significant RNA transcript expression in both linear and ratiometric models in our MPRA: amplicons #3 (overlapping *rs1108075* and *rs11062166*), #6 (overlapping *rs12315711* and *rs2159100*), and #7 (overlapping *rs11062170* and *rs4765905*) (**Figure 4A**). In comparison, no amplicons from the hematocrit LD interval passed significance threshold for activity (**Figure 4—figure supplement 1**). We validated #3 and #6 (highlighted in blue, **Figure 4A**) for enhancer activity in postnatal brain using our single-candidate reporter construct strategy. We also validated lack of activity for #2, an amplicon in the same LD block that did not have significant enhancer activity based on the MPRA results (highlighted in red, **Figure 4A**). Similar to the negative controls and consistent with MPRA findings, #2 did not drive significant EGFP expression in the brains of P7 mice (**Figure 4B**, top panels). On the other hand, #3 and #6 drove detectable EGFP expression in cells throughout the cerebral cortex wherever viral exposure was detected by the co-injected CAG-driven mRuby3 positive control (**Figure 4B**, middle and bottom panels respectively). In a follow-up experiment, we transduced scAAV9-HspMinP-EGFP-#3 and AAV9-CAG-mRuby3 at P0 but waited to collect the brains until P28, at which time we observed that amplicon #3 drove EGFP expression in cortical neurons of adolescent mice (**Figure 4C**, **Figure 4—figure supplement 2**).

These results suggest that the AAV MPRA implementation was effective at screening putative regulatory sequences for in vivo activity in the brain, with reporter expression concordant between MPRA results and single candidate tests of EGFP expression in P7 mouse forebrain for five individually validated amplicons, and that these results could be extended to study activity in later development.

## Discussion

The ability to test in parallel the regulatory capacity of candidate sequences in vivo offers considerable potential for elucidating the role of enhancers in the developing brain and for efficient identification of enhancers that are capable of driving precise expression patterns. Here, we report successful rAAV-mediated delivery of a 3'-oriented parallelized enhancer reporter assay to early postnatal mouse brain and demonstrate the utility of this approach in screening human DNA sequences for regulatory activity in the brain. Via this MPRA, we identified novel presumed enhancers active in P7 mouse cortex, showcasing example applications including identifying regulatory sequences associated with

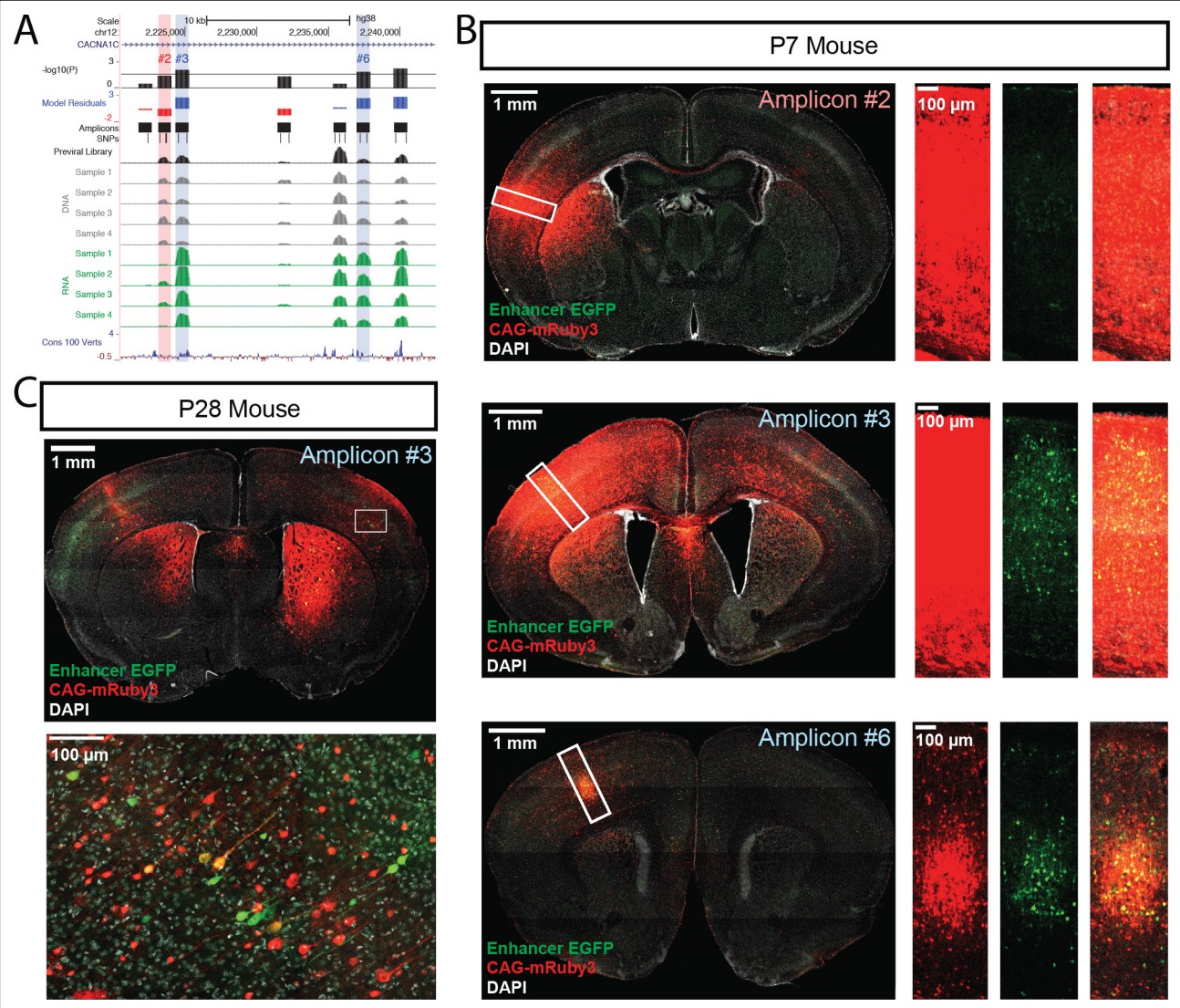

**Figure 4.** Functional dissection of the large third intron of *CACNA1C*. (**A**) UCSC Genome Browser representation of amplicons #1 through #7 in the third intron of *CACNA1C* (hg38, chr12:2,220,500–2,242,499). UCSC tracks for GENCODE v36 and 100 vertebrate conservation, normalized coverage of aligned reads for the previral library, and DNA and RNA samples for the four biological replicates are shown; y-axis scale is 0–50,000 reads. MPRA analysis is shown as graphs of linear model residuals and -log$_{10}$ transformed p-values. Three amplicons, #3, #6, and #7, were found significantly active in our assay. Amplicons which were tested in single-candidate experiments are highlighted (red for no activity in MPRA, blue for significant activity in MPRA) (**B**) Confocal images of single-candidate validation of amplicons #2, #3, and #6. Mice were transduced at P0 with two AAV vectors: one for an HspMinP-EGFP-3'-UTR enhancer reporter construct carrying the indicated amplicon and a second control vector, CAG-mRuby3. Brains were fixed at P7 and sectioned and stained with an antibody for EGFP for signal amplification. Tiled, whole section images are shown on the left. Closeup of boxed regions are shown in the panels on the right. Green, EGFP; red mRuby3; grey, DAPI. These experiments validated robust EGFP expression driven by the two positive MPRA hits (#3 and #6), with substantial EGFP reduction for the MPRA negative amplicon #2. (**C**) Mice were transduced with AAV including positive amplicon #3 and processed as in B, but were raised to P28 before fixing, sectioning, and staining.

The online version of this article includes the following figure supplement(s) for figure 4:

**Figure supplement 1.** Functional dissection of the third intron of *CACNA1C*.

**Figure supplement 2.** Amplicon #3 in the large intron of *CACNA1C* continues to drive EGFP expression in mouse brain at P28.

**Figure supplement 3.** Replicates for functional dissection of *CACNA1C* LD interval.

ASD-associated loci, screening amplicons that include lead SNPs from human genetic studies, and comprehensive testing for enhancers harboring SNPs from disease-associated non-coding LD intervals. We show that amplicons active in our MPRA were more likely to have enhancer signature across functional genomics datasets and that orthologous mouse sequences tested in an independent

parallel reporter assay showed strong activity correlation. Finally, we validated MPRA activity predictions via imaging of EGFP for four positive and two negative sequences in vivo in P7 brain, and we confirmed that activity for one of these positives continued to P28. This study provides a model for applying this powerful screening approach in vivo in mammalian brain.

Our study represents one of the first parallelized enhancer assays testing human sequences in early postnatal mouse brain via rAAV-mediated delivery. We validated MPRA performance, verifying both the capacity for the construct to drive characteristic cell-type restricted expression for a known interneuron enhancer, as well as demonstrating EGFP protein expression driven by novel putative enhancers that were active in vivo in P7 mouse cortex. Based on MPRA and validation experiments, the main sources of variation across MPRA replicates were rAAV transduction rate and injection site (*Figure 1—figure supplement 4*, *Figure 3—figure supplement 2*, *Figure 4—figure supplement 3*), highlighting the need for delivery controls to ensure reproducibility when applying MPRAs in vivo via viral delivery. We observed some variability in amplicon RNA dropout across replicates, likely due to a combination of transduction efficiency variability across animals, sensitivity recapturing amplicons with low viral representation, and PCR stochasticity (*Kebschull and Zador, 2015*). Based on the high degree of correlation between biological replicate #4 and its high amplification cycle technical replicate, we do not believe that PCR amplification bias explains this replicate-specific dropout. On the other hand, we consider it likely that stochastic differences in the number and population of transduced cells in each replicate would alter the apparent activity of amplicons in the assay. For this reason, it is critical that AAV MPRAs use high titer preparations and take advantage of serotypes with high infectivity for the target tissue to maximize the number and diversity of cell types sampled. While not an obvious issue here, the effect of PCR-based clonal amplification bias in other MPRAs has been shown to be reduced with the addition of barcodes and unique molecular identifiers to the reporter construct (*Neumayr et al., 2019*).

Our rAAV-based approach enabled efficient transduction of the mammalian central nervous system. Our approach differs from recent MPRA strategies which use lentiviral vectors (*Klein et al., 2020*; *Gordon et al., 2020*; *Morgan et al., 2020*; *Inoue et al., 2017*) in that AAV is expressed episomally while lentivirus must first integrate into the genome. Lentivirus is especially useful for in vitro applications with difficult to transfect cell types, as AAV is not very efficient at transducing cell lines (*Ellis et al., 2013*). AAV offers a complementary strategy to lentivirus in vivo, as well as considerable flexibility in targeting various tissue types based on capsid serotypes (*Srivastava, 2016*). AAV is gaining traction as a powerful method for enhancer-driven, cell-type-specific manipulations (*Graybuck et al., 2021*; *Mich, 2021*; *Rubin et al., 2020*).

Amplicons in this assay were approximately 900 bp long, representing some of the longer sequences tested in MPRAs to date. Our rationale, that longer sequences afford more native biological context to enhancer activity, appears in accordance with a recent study comparing MPRA designs (*Klein et al., 2020*), which finds that longer sequences add biological signal including enrichment of an RNA polymerase III catalytic subunit, histone-modifying enzymes, and increased transcription factor binding. In our vector design, ~900 bp was the maximum length an amplicon could be and still be efficiently packaged into an scAAV. Using traditional, single-stranded AAV vectors would increase packaging capacity, potentially with a trade-off of reduced transduction efficiency. However, increased length, especially when inserted into the 3'-UTR, may contribute to increased variability across replicates (*Klein et al., 2020*), and may drive mRNA degradation (*Rabani et al., 2017*) that would inhibit RNA transcript detection in the assay. We detected a number of amplicons that had reduced RNA compared to expected background levels, which may represent sequences with silencer or repressor activity such as those that have been reported in recent MPRA screens (*Doni Jayavelu et al., 2020*). However, because of both the insertion of these amplicons into the 3'-UTR and the length of the amplicons, the modified 3'-UTR may have caused transcripts to be subject to RNA degradation. Thus, we are hesitant to draw conclusions about these amplicons with lower than expected activity without further characterization. Nevertheless, while artifacts from inserting sequences into the 3'-UTR may impact the STARR-seq design and our results, we show that orthologous mouse sequences exhibited correlated presence and absence of activity and similarly that amplicons with absence or presence of MPRA activity across replicates consistently reproduced in independent single deliveries to the brain. Further, active amplicons were enriched for DNase Hypersensitivity Sites, H3K4me1, and H3K4me3 peaks generated from fetal human brain tissue, and with ATAC-seq peaks from FACS-purified neurons

and glia from postmortem human brain. Overall, the combination of reproducible MPRA activity, enrichment for neuron-specific enhancer signatures, and activity validation in AAV experiments provide evidence that positive results of the MPRA indeed act as enhancers.

Parallelized enhancer assays such as the one reported here have the possibility to become fundamental for assessing active enhancers in the brain and offer great potential for functional dissection of sequence-based enhancer activity. The vast majority of sequence variants associated with genetic risk for neurodevelopmental and neuropsychiatric disorders are found in non-coding regions (*Zhang and Lupski, 2015*), many of which are presumed to be located in enhancers (*Ripke, 2014*). As an example, our assay enabled functional annotation of 17 intronic *CACNA1C* amplicons spanning regions harboring schizophrenia- and bipolar disorder-associated SNPs within strong linkage disequilibrium and identified three regions with enhancer activity in the brain. Notably, we detected no activity in our assay from amplicon #5 (LD group), which spans a region harboring the SNPs *rs1006737* and *rs2007044*, two of the most statistically significant risk SNPs (*Moon et al., 2018*) from GWAS of schizophrenia and bipolar disorder (*Figure 4A*). Two of the three amplicons we identified as enhancers in P7 brain had prior evidence of enhancer activity in in vitro models (*Eckart et al., 2016*; *Roussos, 2014*), indicating our assay can reproducibly detect enhancer activity identified in orthogonal studies. The region spanning amplicon #6 was previously annotated as a putative enhancer in physical proximity to the *CACNA1C* promoter in human induced pluripotent stem cell (hiPSC)-derived neurons (*Roussos, 2014*). The region spanning amplicon #7 exhibited enhancer activity via luciferase assay in SK-N-SH cells (*Eckart et al., 2016*). In addition, #7 was predicted as a putative enhancer based on open chromatin signatures in both fetal and postmortem human brain (*Dunham, 2012*; *Bernstein et al., 2010*; *Fullard et al., 2018*). Finally, we show that activity of one of these *CACNA1C* intronic enhancers continues at P28, highlighting the potential to apply the MPRA and single enhancer AAV testing at later ages. Although our design was not well powered to compare the activity of sequence variants (*Figure 2—figure supplement 4*), future studies using allele-balanced libraries, libraries with reduced complexity, or libraries with allele-specific barcodes could greatly improve the sensitivity and power of this assay to evaluate the effects of sequence variation on enhancer activity. We attempted to assess allelic differences in enhancer function for amplicon #3 using a luciferase assay, but we found no difference in activity between the reference and variant SNPs. However, disease-relevant SNPs may alter enhancer function by modulating activity in specific cell-type or developmental contexts which were not replicated in the in vitro luciferase assay. For this reason, in vivo parallel functional assays will be critical in understanding how sequence variation within enhancers contributes to altered gene expression, including for the disease-associated *CACNA1C* locus.

In summary, our in vivo parallelized enhancer reporter assay in P7 mouse brain enabled us to identify novel enhancers active in early postnatal mouse brain and pinpoint potential regulatory regions where disease-associated sequence variation (e.g. GWAS SNPs) may contribute to transcriptional pathology. Our results highlight the opportunities gained by in-depth functional dissection of enhancers in the developing brain via in vivo adaption of MPRAs, toward deeper understanding of enhancer biology across neurodevelopment and in the etiology of neuropsychiatric disorders.

# Materials and methods
## Selection of library candidates

For the miniMPRA, 16 noncoding sequences were manually selected based on putative regulatory activity by mouse H3K27ac data (*Nord, 2013*). For the full MPRA, 408 candidates were selected for screening based on the following criteria: 22 regions containing *CACNA1C*-associated SNPs (*rs1006737, rs2007044, rs4765913, rs4765905, and rs7312105*) and variants in linkage disequilibrium ($r^2$ >0.8) of listed SNPs; 70 regions containing *SCN1A*-associated SNPs (*rs7587026, rs12987787, rs11890028, rs6732655, rs11692675*) and variants in linkage disequilibrium ($r^2$ >0.8); 29 epilepsy-associated SNPs (GCST001662: *rs13026414, rs72823592, rs10496964, rs12059546, rs39861, rs2717068, rs771390, rs10030601, rs12720541*; GCST002547: *rs28498976, rs2947349, rs1939012, rs55670112, rs535066, rs111577701*; GCST002141: *rs492146, rs72700966, rs61670327, rs143536437, rs11861787, rs72698613, rs12744221*; GCST000691: *rs346291, rs2601828, rs2172802, rs2841498, rs1490157, rs2475335*; GCST001329: *rs2292096*); 142 regions overlapping schizophrenia-associated SNPs (*Ripke, 2014*); 129 regions overlapping DNaseI hypersensitive sites identified from fetal human

brain (*Dunham, 2012*; *Bernstein et al., 2010*) found within autism-associated copy number variants (*Turner, 2016*; *Turner et al., 2017*); and 16 human homologs of the mouse miniMPRA sequences. Schizophrenia- and epilepsy-associated SNPs were identified using the NHGRI-EBI GWAS Catalog (*Buniello et al., 2019*). SNPs in linkage disequilibrium for query SNPs listed as *CACNA1C-* or *SCN1A-*associated above were identified using HaploReg v3 (*Ward and Kellis, 2012*) using LD reported from the 1000 Genomes Project (*Auton et al., 2015*). Full selection metadata for each region of interest can be found in *Supplementary files 1 and 2*.

## Vector preparation and library cloning

We replaced the CMV-EGFP expression cassette in scAAV-CMV-EGFP (*Rosas et al., 2012*) with a synthesized gBlocks Gene Fragment (IDT) containing the *Hsp1a* minimal promoter followed by the gene for EGFP with a cloning site inserted into the 3'-UTR. This cloning site consisted of two restriction sites flanked by two 35 bp sequences used for Gibson homology. The resulting plasmid, pscAAV-HspMinP-EGFP, housed the 3'-UTR-oriented enhancer reporter construct between the ITR sequences necessary for packaging scAAV particles (*Figure 1—figure supplement 1*). Unlike other STARR-seq constructs (*Arnold et al., 2013*; *Muerdter et al., 2018*; *Lea et al., 2018*), we did not include an intronic sequence in the reporter ORF, as we preferred to reduce construct length and complexity considering the novel implementation in an scAAV context and 900 bp amplicon sequence.

We designed 16 PCR amplicons (miniMPRA) and 408 PCR amplicons (MPRA) covering the regions of interest described above by customizing a batch primer design pipeline using Primer3 (*Untergasser et al., 2012*; *Supplementary files 1 and 2*). PCR primers for the miniMPRA were designed manually using the mouse genome assembly mm10. All PCR primers for the full MPRA were designed using the human genome assembly GRCh37 and the Primer3 human mispriming library *cat_humrep_and_simple.fa*. In brief, amplicons were designed to fit the following criteria with all other parameters set to the Primer3 default: amplicon product size between 880–940 bases; primer size between 18–25 bases with an optimum of 20 bases; primer anneal temperature between 57 °C–61 °C, with an optimum of 60 °C; maximum primer pair annealing temperature difference of 2 °C; and primer GC content between 30 %–70 %. We excluded any primer pairs that contained a SNP within 50 bases of either end of the PCR product. PCR primers were then appended with homology sequences (forward primer: tgtacaagtaacatagtcatttctagaTTAATTAA; reverse primer: gctctagtcgacggtatcgataagcttGGCGCGCC) for Gibson Assembly cloning, up to a total primer length of 60 bases.

To generate the amplicon library, we performed individual PCRs using Phusion Hot Start II High Fidelity DNA Polymerase (Invitrogen #F549S) in 10 µL reaction volumes in 96-well plates with 50 ng of pooled human DNA (1:1 mix of Coriell DNA pool #NA13405 and Coriell DNA pool #NA16600) as the DNA template. Thirty cycles of PCR were performed with annealing at 60 °C and 30 s extension time. We then quantified the concentrations of each PCR reaction using Quant-iT PicoGreen dsDNA Assay Kit (Invitrogen #P7589) to calculate equimolar amounts of each PCR product for downstream pooling.

We next linearized the vector pscAAV-HspMinP-EGFP (*Figure 1—figure supplement 1*) using PacI (NEB #R0547L) and AscI (NEB #R0558L). AAV DNA vectors are prone to recombination at the ITR sites, so we also performed a separate restriction digest using SmaI (NEB #R0141L) using pscAAV-HspMinP-EGFP to check for vector integrity. The PacI/AscI digested vector was run on a 1.0 % agarose gel, excised, and purified using the Promega Gel & PCR Cleanup system (#A9282) and quantified using Nanodrop. After concentration quantification of both vector and PCR products, 5 ng of each PCR product was pooled into one tube for purification using the QIAquick PCR purification kit (Qiagen #28106) and eluted into 30 µL Nuclease Free Water. Pooled DNA concentration and quality was then assessed by Nanodrop and Qubit dsDNA High Sensitivity assay kit (Thermo Fisher #Q33230). We also ran the pooled PCR products on a 1.0 % agarose gel to verify the expected product sizes (~1–1.1 kb).

We performed two separate Gibson assemblies (NEB #E2611S) using 100 ng of PacI/AscI-linearized vector and 3-fold molar excess of pooled PCR amplicons in a 20 µL reaction volume. We then incubated the Gibson assembly mixtures for 1 hour at 50 °C. For bacterial transformation, 2.0 µL of Gibson assembly reaction was added to 50.0 µL of NEB Stable competent cells (#C3040I), in triplicate for each reaction. Competent cells were heat shocked at 42 °C for 30 s, and recovered in 950 µL of SOC media for 1 hour at 37 °C. A total of 100 µL of each transformation replicate was plated onto 100 µg/mL carbenicillin agar plates and grown overnight at 30 °C. 900 µL of cells were transferred directly into 250–500 mL of 100 µg/mL carbenicillin Luria-Bertani broth and grown overnight on a 30 °C shaker

at >250 rpm. Agar plates were compared for colony growth, indicating Gibson assembly success. Individual liquid cultures were purified using the Macherey-Nagel Xtra Maxi EF kit (#740424), eluted into 500 μL Nuclease Free Water, and assessed for concentration and purity using Nanodrop. Randomly sampled individual colonies as well as all maxiprep replicates were assessed by Sanger sequencing to confirm assembly at the cloning site within pscAAV-HspMinP-EGFP, then maxiprep replicates were pooled 1:1 by concentration for the final library used for viral packaging.

## Viral packaging

For both the miniMPRA and the full AAV MPRA, adeno-associated virus (AAV9(2YF)) (*Dalkara et al., 2012*) was produced using a triple plasmid transfection method in 293T cells (*Grieger et al., 2006*). After ultracentrifugation, the interphase between the 54 % and 40 % iodixanol fraction, and the lower three-quarters of the 40 % iodixanol fraction, were extracted and diluted with an equal volume of phosphate-buffered saline (PBS) plus 0.001 % Tween 20. Amicon Ultra-15 centrifugal filter units (Millipore, Bedford, MA) were preincubated with 5 % Tween in PBS and washed with PBS plus 0.001 % Tween. The diluted iodixanol fractions were buffer-exchanged and concentrated to 250 μL in these filter units. Virus was washed three times with 15 mL of sterile PBS plus 0.001 % Tween. Vector was then titered for DNase-resistant vector genomes by real-time PCR relative to a standard.

## Injections of library viruses to neonatal mice

C57BL/6 mice from Jackson Laboratories were used for mouse experiments. All procedures were performed in accordance with the ARVO statement for the Use of Animals in Ophthalmic and Vision Research and were approved by the University of California Animal Care and Use Committee (AUP #R200-0913BC). Surgery was performed under anesthesia, and all efforts were made to minimize suffering. Neonatal mice (on a C57BL/6 J background) were injected in the prefrontal cortex in two separate locations with 0.5 μL of virus. Only the left side of the brain was injected. A beveled 34-gauge needle (Hamilton syringe) was used to push through the skull and to inject virus into the brain. Virus was delivered slowly by hand over the course of 1 min. Mice were anesthetized on ice for 3–4 min prior to injection, and after injection mice were heated on a warm pad before being returned to the cage.

## Tissue collection, library preparation, and sequencing

Bulk forebrains were dissected from P7 mice injected with the viral library at P0. Dissection included whole forebrain, separated into left and right hemispheres, after removing surface tissue and skull. Hemispheres were stored in RNAlater Stabilization Solution (Thermo Fisher #AM7020) for downstream dual DNA and RNA collection. Hemispheres were then physically homogenized and processed using the AllPrep DNA/RNA Mini Kit (Qiagen #80204) to collect total RNA and genomic DNA from the same tissue. RNA was treated on-column with 1 μL RNase-Free DNase (Qiagen #79254). First strand cDNA was synthesized from purified total RNA using the SuperScript VILO cDNA Synthesis Kit (Invitrogen #11755050) with random primers. We amplified the variable regions of the reporter library from cDNA and genomic (g)DNA using PCR with Phusion Hot Start II High Fidelity DNA polymerase (Invitrogen #F549S) using primers flanking the variable region within the 3'-UTR of pscAAV-HspMinP-EGFP (forward primer, Amplicon PCR Primer F: GATCACTCTCGGCATGGAC; reverse primer, Amplicon PCR Primer R: GATGGCTGGCAACTAGAAGG). PCR consisted of 30 cycles (or 35, for technical replicate Sample 4–35) at 60 °C for annealing and 1 min for extension. PCRs were then loaded onto a 1.0 % agarose gel, the ~1–1.1 kb fragment excised, and gel fragment purified using the Promega Gel & PCR Cleanup system (#A9282). Purified gel fragments were then re-purified using the Qiagen MinElute PCR Purification Kit (#28004) to remove residual chemical contamination, and concentration and quality assessed with Nanodrop and the Qubit dsDNA High Sensitivity assay kit (Thermo Fisher #Q33230). Purified amplicons were prepared for sequencing using the Nextera XT DNA Library Preparation Kit (Illumina #FC-131–1001) using 1 ng of purified amplicon library per sample. Quality and concentration of sequencing libraries was assessed using an Agilent BioAnalyzer instrument. Each library was quantified and pooled for sequencing on the Illumina HiSeq 4000 using a paired end 150 bp sequencing strategy (*Supplementary file 8*). The miniMPRA was sequenced on a MiSeq instrument since lower coverage was required due to lower library complexity.

## Sequencing alignment

Raw sequencing reads were trimmed to remove Nextera XT adapters using NGmerge (v0.2_dev) (*Gaspar, 2018*) with maximum input quality score set to 41 (-u 41). Trimmed reads were aligned to the human genome (GRCh38) using BWA-MEM (v0.7.17) (*Li, 2013*) with an option marking shorter split hits as secondary (-M) and otherwise standard parameters. As additional quality control, reads were also aligned to the mouse genome (GRCm38) to check for cross-contamination of mouse DNA from the amplicon PCR. Aligned SAM files were processed using Picard tools (v 2.23.3) (*Eckart et al., 2016*): aligned reads were sorted by coordinates and converted to BAM files using *SortSam*, optical duplicates were flagged by *MarkDuplicates*, and BAM file indexing was conducted using *BuildBamIndex*. The number of de-duplicated reads within in silico PCR amplicon coordinates was counted using *samtools view -F 0 × 400* c (v1.7) (*Li et al., 2009*) and reported as the raw amplicon count.

## Allele-specific MPRA activity

Allele-specific counts were generated from deduplicated bam files using bam-readcount (v0.8.0) (https://github.com/genome/bam-readcount; *Khanna et al., 2021*) with min-base-quality set to 15, and min-mapping-quality set to 20. Allelic counts were further processed using a custom R script, applying min base quality filter of 25, and min allele frequency filter of 0.01, producing a set of 440 biallelic SNPs.

## Bioinformatics analysis with multiple linear regression

Analyses were performed in R (v4.0.2) and RStudio (v1.3.1093) software. To consider an amplicon as present in our MPRA library, we set a threshold of at least 200 read counts per amplicon in our library sample, resulting in a reduced dataset from 408 to 345 amplicons. This threshold was established based on the distribution of library counts. No GC or sequence bias was detected following *in silico* PCR analysis when present and missing amplicons were compared. Two primer pairs, expected to produce PCR products in our original design using the hg19 reference genome had no predicted products in our *in silico* approach, as well as in our MPRA library aligned to the GRCh38 genome (*Supplementary file 9*). *In silico* PCR analysis was performed using a custom R script and DECIPHER R package (*Erik, 2016*).

Raw amplicon counts were converted to proportion by normalizing raw count values for each amplicon to all amplicon counts in a given sample. A value of 1 was added to the numerator and denominator to avoid dividing by zero. For downstream analysis, mean amplicon proportions for genomic DNA ('DNA') and cDNA ('RNA') were calculated across all samples excluding the technical replicate Sample 4–35. RNA/DNA ratio ('ratiometric activity') was calculated by dividing porportion$_{RNA}$ by proportion$_{DNA}$ for each amplicon per replicate. After proportion conversion, the dataset was cleaned by removing low count amplicons with less than 200 raw counts in any of the DNA samples, and mean amplicon library proportions less than $2^{-15}$, resulting in 308 amplicons for downstream analysis.

The dataset used for building the linear model was selected by first removing the 10 % of amplicons with the highest and 10 % with the lowest mean ratiometric activity across the four biological replicates, resulting in 247 amplicons used for training the model. The model was fit using log$_2$-transformed mean RNA and log$_2$-transformed mean DNA proportions, with amplicon GC content provided as a model covariate. GC content had a significant impact, but low effect size, on the model as demonstrated by higher P values, $1.416 \times 10^{-8}$ vs $< 2.2 \times 10^{-16}$, and lower F statistics, 34.446 vs 659.44, when log$_2$ DNA proportions were compared with the GC content. Inclusion of the GC content was justified based on the reduction of Bayesian information criterion (BIC), from 626.2976 to 599.1863, for models without and with GC content, respectively. The normality of residuals distribution was confirmed using the Kolmogorov-Smirnov test and a graphical approach. The model was applied to the complete dataset, and one-tailed p-values were calculated from the Z-scaled distribution of residuals, using the normal distribution implemented using R stats pnorm (*R Development Core Team, 2018*). Empirical tail area-based FDR q-values, estimated from Z scores, were calculated using fdrtool R package (*Strimmer, 2008*), with threshold adjustment for one-tail testing. We compared the linear activity model to using ratiometric activity alone. To control low-count variability, 1000 counts were added to RNA and DNA counts to stabilize ratios where counts were low. Mean ratiometric activity and standard deviation was then calculated for each amplicon across all biological replicates, excluding the technical replicate Sample 4–35. We compared the mean ratio, mean ratio – 1. s.d., ranked residuals,

and p-value for all amplicons passing initial quality control. We also compared our linear model with a Wilcoxon rank sum, a non-parametric test comparing DNA vs RNA proportions, using Benjamini-Hochberg FDR < 0.05 for multiple testing correction.

## Intersections with epigenomics datasets

Amplicons were intersected with epigenomics datasets from Roadmap (macs2 peaks filtered at q < 0.05) (*Bernstein et al., 2010*), with BOCA merged neuron and glia ATAC-seq peaks from post-mortem human brain tissues (*Fullard et al., 2018*), with DNase I digestion-based digital TF footprints (*Vierstra et al., 2020*), and with vertebrate evolutionary conserved elements in the human genome from the UCSC Genome Browser. Consolidated peak intervals (genomic ranges) were merged prior to intersecting using R package GenomicRanges reduce function. Intersections were carried out using GenomicRanges findOverlaps function. Significance of enrichment was calculated using a one-tailed permutation test with 20,000 random samplings. The initial selection bias for active amplicons was accounted for by using the background of our MPRA library for random sampling, ensuring the robustness of statistical testing.

The predictive power of epigenomic signatures for MPRA activity was tested by adding the dichotomous Boolean variables of epigenomic intersection to a general linear model (GLM) predicting the level of RNA from DNA, and a GC content covariate, one epigenomic intersection at a time using the R stats glm function. T-test p-values were extracted from model summary, and models with and without the epigenomic covariates were compared using BIC. In addition to the GLM, we calculated point-biserial correlations between the Z-scaled residuals from our original lm model, and each epigenomic signature intersection using the biserial.cor function from the ltm R package (*Rizopoulos, 2006*).

## Single-candidate enhancer reporter cloning

To validate selected positive hits, single-candidate enhancer reporter plasmids were constructed. Candidate sequences, amplicons #2, #3, #6 (LD group), #161 (FBDHS group), or #264 (PutEnh group negative control) were cloned by PCR from a pooled sample of human DNA (1:1 mix of Coriell DNA pool #NA13405 and Coriell DNA pool #NA16600) using the same primer sequences as used to clone the amplicons in the combined test library. Each amplicon was then inserted between the PacI/AscI cloning site downstream of EGFP in pscAAV-HspMinP-EGFP, using In-Fusion (Takara Bio #639649) or Gibson Assembly (NEB #E2611S) according to manufacturer's specifications. In-Fusion or Gibson reaction products were used to transform Stellar competent cells (Takara Bio #636763) via heat shock at 42 °C and ampicillin-resistant clones were selected at 37 °C using LB-agar plates inoculated with carbenicillin. Clones were confirmed by PCR and Sanger sequencing. Amplicon #2 included the following alleles: A at *rs4765904*, A at *rs1108221*, and C at *rs1108222*. Amplicon #3 included the following alleles: G at *rs1108075* and G at *rs11062166*. Amplicon #6 included the following alleles: C at *rs2159100* and T at *rs12315711*. The integrity of the viral ITR sequences was verified by restriction digest with XmaI (NEB #R0180L) before proceeding with AAV packaging.

## Single-candidate enhancer reporter AAV packaging

Adeno-associated viruses for single-candidate enhancer reporter constructs were packaged in-house using a triple-transfection protocol followed by concentration of viral particles from culture media using a table-top centrifuge protocol (*Broussard et al., 2018*). Briefly, for each construct, ~$10^6$ AAV293 cells (Agilent #240073) were plated on a 10 cm tissue culture dish in 10 mL of standard DMEM media (Corning #10–013-CV) supplemented with 10 % FBS (ThermoFisher #10082139) and 1 % penicillin/streptomycin (ThermoFisher #15070063) and maintained in a humidified incubator at 37 °C and 5 % $CO_2$. After cells reached ~60 % confluency (1–2 days), the cells were passaged and split onto two 10 cm dishes and grown for an additional 1–2 days before passaging again onto two 15 cm dishes in 20 mL of supplemented culture media each. When the cells in 15 cm dishes were 60–80 % confluent the triple-transfection protocol was performed. The media was changed for fresh, supplemented DMEM 1 hr before transfection and the culture was returned to the incubator. 17 µg of enhancer reporter plasmid (or CAG-driven mRuby3 control plasmid) was mixed with 14.5 µg AAV helper plasmid (gift from Tian lab), 8.4 µg AAV9 rep/cap plasmid (gift from Tian lab), and 2 mL jetPRIME buffer (Polyplus #114–15), vortexed 10 s and incubated at room temperature for 5 min. Eighty µL

jetPRIME transfection reagent (Polyplus #114–15) was then added to the plasmid/buffer mixture, vortexed, and incubated at room temperature for 10 min. One mL of jetPRIME transfection solution was added dropwise to each 15 cm culture dish which was then returned to the incubator overnight. The next day, the transfection media was exchanged for fresh, supplemented DMEM, and the cultures were incubated for an additional 2 days to allow viral particles to accumulate in the media. Approximately 3 days post-transfection, ~40 mL of media from both 15 cm culture plates was combined in a 50 mL conical centrifuge tube. Cellular debris was pelleted using a table-top centrifuge at 1500 x g for 1 min, and the supernatant was filtered over a 0.22 µm filter (EMD Millipore #SLGP033RS) into a fresh conical tube. The viral particles were precipitated from the media using AAVanced concentration reagent (System Biosciences #AAV100A-1) according to manufacturer's instructions. Briefly, 10 mL of pre-chilled AAVanced was added to 40 mL filtered media, mixed by pipetting, and then incubated at 4 °C for 24–72 hr. After incubation, viral particles were pelleted by centrifugation at 1500 x g for 30 min at 4 °C. The supernatant was removed, the viral pellet was resuspended in 500 µL cold, serum-free DMEM, transferred to a 1.5 mL Eppendorf tube, and centrifuged again at 1500 x g for 3–7 min. At this step, if there was an AAVanced/viral pellet the supernatant was removed by aspiration and the pellet was resuspended in 40–60 µL cold Dulbecco's PBS (Fisher Scientific #MT21031CV). If there was no viral pellet after this final centrifugation, we used the viral DMEM solution as our final virus prep. The virus solution was then aliquoted, flash frozen on liquid nitrogen, and stored at –80 °C until use. Once thawed, an aliquot was stored at 4 °C.

Viral titer was estimated by qPCR with an MGB-NFQ FAM-labeled probe that recognized GFP (Life Technologies, sequence: 5′-CTTCAAGGAGGACGGCAA-3′) and using pscAAV-HspMinP-EGFP vector plasmid as a standard. DNA was purified from 1 µL of virus preparation and 20 ng of plasmid DNA using the QIAGEN DNeasy Blood and Tissue kit (QIAGEN #69504). qPCR reactions of serial-dilutions of the DNA samples and plasmid standards were prepared using TaqMan Universal PCR Master Mix (ThermoFisher #4304437) in a 384-well plate and run on an Applied Biosystems QuantStudio Real-time PCR instrument at the UC Davis School of Veterinary Medicine Real-time PCR Core Facility. Viral titers ranged from $6.8 \times 10^{10}$–$2.6 \times 10^{12}$ genome copies/mL, and all viruses were normalized to the lowest titer for each experiment.

## Injections of single-candidate enhancer reporter viruses to neonatal mice

Intracranial virus injections were performed on C57BL/6 mice at P0-1 using protocols approved by the UC Davis Institutional Animal Care and Use Committee (Protocol #20506). First, mouse pups were cryo-anesthetized until cessation of pedal withdrawal reflex. Each pup was then injected in each lateral ventricle with 1 µL of a virus mixture consisting of a 2:1 ratio of enhancer reporter virus to CAG-mRuby3 control virus and containing 0.06 % Fast Green dye (Grainger #F0099-1G) (2 µL total, ~1.4 x $10^8$ particles of enhancer reporter virus). The injection sites for each ventricle were located midway between anatomical landmarks lambda and bregma, approximately midway between the eye and the sagittal suture. After injection, the mouse pup was placed in a warmed recovery chamber for 5–10 min before being returned to its parent's home cage.

## Immunohistochemistry

After 7 or 28 days of virus incubation, the mice were anesthetized under isoflurane and sacrificed by transcardial perfusion with 4 % PFA in PBS. The brain was then dissected and placed in 4 % PFA overnight to complete fixation. The fixed brain was dehydrated in a 30 % sucrose PBS solution for 3–4 days before freezing in O.C.T. medium (VWR #25608–930). 30 to 35 µm coronal sections were cut on a cryostat and collected in PBS. Floating sections were stained in 24-well plates using the following protocol. All steps at 4 °C and room temperature were performed with gentle agitation on an orbital shaker (Cole-Parmer product #60–100 RPM). Antigen retrieval was performed in 1 x Citrate buffer (pH 6.0; Sigma-Aldrich #C9999-1000ML) at 60 °C for 1 h. After allowing to cool for 20 min, sections were permeabilized for 20 min at room temperature in PBS containing 0.5 % Triton X-100. Sections were then blocked for 1 h at room temperature in a blocking solution containing 0.1 % Triton X-100 and 5 % milk in PBS. The sections were then transferred to blocking solution containing primary antibodies and incubated for 1–3 days at 4 °C. After primary antibody incubation, sections were washed 5 times in 0.1 % Triton X-100 PBS for 20 min each time. Sections were then transferred to blocking solution

containing secondary antibodies and incubated for 45 min to 1 hr at room temperature, followed by 3–5 more 20 min washes in 0.1 % Triton X-100. Finally, sections were stained with DAPI (ThermoFisher #D1306) at room temperature for 20–30 min and mounted on slides with ProLong Gold Antifade Mountant (ThermoFisher #P36934). Primary antibodies included chicken anti-GFP (ThermoFisher #A10262, 1:1000) and rat anti-Ctip2 (Abcam #ab18465, 1:500), and secondary antibodies included a Donkey anti-chicken AlexaFluor-488 (Jackson ImmunoResearch #703-545-155, 1:500) and a Donkey anti-rat AlexaFluor-647 (Jackson ImmunoResearch #712-605-153, 1:500 or 1:250).

For immuno-fluorescent co-staining of EGFP and Lhx6, a slightly different staining protocol was used. Thirty μm coronal brain sections were first mounted onto microscope slides and then incubated at 37 °C for 30 min. Next, the slides were treated to a series of steps to perform antigen retrieval to detect the Lhx6 antigen. These included 8 min shaking in a base solution of 0.5 g NaOH in 50 mL water, followed by 8 min shaking in 0.3 % glycine in 1 x PBS, and finally 8 min shaking in 0.3 % SDS in 1 x PBS. The slides were then incubated in a blocking solution of 5 % BSA and 0.3 % Triton X-100 in 1 x PBS for 1 h, followed by primary antibody incubation overnight at 4 °C. The next day, slides were washed in 0.3 % Triton X-100 in 1 x PBS three times and then incubated in secondary antibodies for 1 h at room temperature. Slides were finally washed three more times and coverslips mounted with Vecta-shield anti-fade medium (Vector labs, H-1000). Primary antibodies included a mouse anti-Lhx6 (Santa Cruz Biotechnologies #sc-271443, 1:200) and a rabbit anti-GFP (ThermoFisher #A11122, 1:2000), and secondary antibodies included a goat anti-mouse AlexaFluor-647 (ThermoFisher #A32728, 1:300) and a goat anti-rabbit AlexaFluor-488 (ThermoFisher #A32731, 1:300).

## Fluorescence microscopy and image analysis

Confocal images were acquired on a Zeiss LSM800 microscope using a 0.25 NA 5 x air objective or a 0.8 NA 25 x oil objective. Images were acquired using the same settings for laser intensity, PMT gain, and pixel dwell time. For whole-brain section images showing overall injection and virus expression, images were tiled across the section with a 10 % overlap and stitched in postprocessing using the Stitching plugin in Fiji64 (*Preibisch et al., 2009*). Brain sections co-labeled for Lhx6 were imaged using a Leica DM2000 fluorescent compound microscope with a mounted DFC3000 G mono-chrome camera. Images were acquired using a 10 x objective which was sufficient to capture the entire neocortical region from white matter to pia. For all images, cells expressing EGFP or mRuby3 were counted and analyzed using Fiji (*Schindelin et al., 2012*). Images with co-labeling were then scored for EGFP and mRuby3 cells that were double-positive for the co-label. Example images shown have been processed with a 3 × 3 median filter in Fiji, background has been subtracted from each channel, and the contrast of each channel has been adjusted to the same levels for comparison across images.

## Luciferase assay

We constructed luciferase reporter plasmids by replacing the minP promoter in pGL4.24 (Promega) with the *Hsp1a* minimal promoter using restriction enzymes HindIII and NcoI to make pGL4.24-HspMinP. We then cloned a ~900 bp region containing either Amplicon #256, a predicted nega-tive (forward primer: 5'- ccggagctcTGGTATGGGTGAAAACGGCT-3'; reverse primer: 5'- cggctcgag GAGGTTTGTGGGGAGGAGTG-3'), or Amplicon #3 (forward primer: 5'- ttaattaaggtaccTCTAGAGC TCTCTTTTGAACTGC-3'; reverse primer: 5'- ttaattaactcgagATAATTCTCTTTGTGCAATGCTACA-3') into the pGL4.24-HspMinP vector (Promega) upstream of HspMinP using restriction enzymes XhoI and either SacI or KpnI. Sequence validation of the plasmids allowed us to distinguish #3 reference and variant. HEK293 cells and SH-SY5Y cells were obtained from ATCC and cultured following stan-dard protocols. No further validation or testing of cell line veracity or contamination was performed. When cultures were 40–60 % confluent, cells were transfected using Lipofectamine 3000 (Invitrogen) with each construct (400 ng) and the Renilla luciferase expression vector pRL-TK (40 ng; Promega) in triplicate. After 24 h, the luciferase activity in the cell lysates was determined using the Dual Luciferase Reporter System (Promega). Firefly luciferase activities were normalized to that of Renilla luciferase, and expression relative to the activity of the empty pGL4.24-HspMinP vector was noted.

## Acknowledgements

Sequencing was performed at the UC San Francisco and UC Davis DNA cores. We thank John Flan-nery for advice and providing space in which we performed preliminary experiments. We also thank

the lab of Lin Tian at UC Davis for generously gifting us AAV helper and rep/cap plasmids. This work was supported by NIH/NIGMS R35GM119831. L.S.-F. was supported by the UC Davis Floyd and Mary Schwall Fellowship in Medical Research, the UC Davis Emmy Werner and Stanley Jacobsen Fellowship, and by grant number T32-GM008799 from NIGMS-NIH.

## Additional information

### Funding

| Funder | Grant reference number | Author |
| --- | --- | --- |
| National Institutes of Health | R35GM119831 | Alex S Nord |
| National Institutes of Health | T32-GM008799 | Linda Su-Feher |

The funders had no role in study design, data collection and interpretation, or the decision to submit the work for publication.

### Author contributions

Jason T Lambert, Conceptualization, Formal analysis, Investigation, Methodology, Project administration, Validation, Visualization, Writing – original draft, Writing – review and editing; Linda Su-Feher, Conceptualization, Formal analysis, Investigation, Methodology, Project administration, Visualization, Writing – original draft, Writing – review and editing; Karol Cichewicz, Data curation, Formal analysis, Investigation, Methodology, Software, Visualization, Writing – original draft, Writing – review and editing; Tracy L Warren, Formal analysis, Investigation, Visualization, Writing – review and editing; Iva Zdilar, Investigation, Methodology; Yurong Wang, Data curation, Investigation, Methodology, Software, Writing – review and editing; Kenneth J Lim, Software; Jessica L Haigh, Daniel Vogt, Formal analysis, Investigation, Methodology, Validation, Writing – original draft, Writing – review and editing; Sarah J Morse, Tyler W Stradleigh, Investigation, Methodology, Validation; Cesar P Canales, Formal analysis, Investigation, Methodology, Validation; Erika Castillo Palacios, Viktoria Haghani, Spencer D Moss, Hannah Parolini, Diana Quintero, Diwash Shrestha, Formal analysis, Investigation, Validation; Leah C Byrne, Investigation, Methodology, Validation, Writing – original draft, Writing – review and editing; Alex S Nord, Conceptualization, Data curation, Funding acquisition, Project administration, Resources, Supervision, Writing – original draft, Writing – review and editing

### Author ORCIDs

Jason T Lambert ⓘ http://orcid.org/0000-0002-0264-9482
Linda Su-Feher ⓘ http://orcid.org/0000-0002-0660-7925
Karol Cichewicz ⓘ http://orcid.org/0000-0001-5926-3663
Tracy L Warren ⓘ http://orcid.org/0000-0001-5125-0868
Iva Zdilar ⓘ http://orcid.org/0000-0002-0080-3132
Jessica L Haigh ⓘ http://orcid.org/0000-0002-9518-4003
Cesar P Canales ⓘ http://orcid.org/0000-0003-2505-8367
Viktoria Haghani ⓘ http://orcid.org/0000-0002-3700-4027
Alex S Nord ⓘ http://orcid.org/0000-0003-4259-7514

### Ethics

All procedures were performed in accordance with the ARVO statement for the Use of Animals in Ophthalmic and Vision Research and were approved by the University of California Animal Care and Use Committee (AUP #R200-0913BC). Surgery was performed under anesthesia, and all efforts were made to minimize suffering.

### Decision letter and Author response

Decision letter https://doi.org/10.7554/eLife.69479.sa1
Author response https://doi.org/10.7554/eLife.69479.sa2

## Additional files

**Supplementary files**
• Supplementary file 1. miniMPRA_counts_prop_act.csv. Chr, Start_mm10, End_mm10 – Genomic coordinates of the miniMPRA amplicons;miniMPRA_ID – miniMPRA identifiers;Maxi, S1, S2, S3, S4 – pre-viral library Maxiprep, and RNA library counts of 4 biological replicates;miniMPRA_mean_counts, miniMPRA_SD_counts – mean and standard deviation of sample S1-S4 counts;Maxi_prop, S1_prop, S2_prop, S3_prop, S4_prop – proportions of counts normalized to library depth;miniMPRA_mean_counts_prop, miniMPRA_SD_counts_prop – miniMPRA_mean_counts miniMPRA_SD_counts scaled by their count proportions;miniMPRA_mean_prop, miniMPRA_SD_prop – mean and standard deviation of S1-S4_prop;S1_act, S2_act, S3_act, S4_act – MPRA activity calculated relatively to the Maxi_prop; miniMPRA_mean_act, miniMPRA_SD_act – mean and standard deviation of S1-S4_act;ID – Amplicon number of the orthologous MPRA amplicon in the STAR408 library.

• Supplementary file 2. STAR408_bed_GRCh38_coordinates.csv. Chr, Start_GRCh38, End_GRCh38 – genomic coordinates of MPRA amplicons;Amp_name – amplicon name, including amplicon number;Group – Amplicon group: LD, GWAS, FBDHS, or PutEnh.

• Supplementary file 3. STAR408_counts.csv. Chr, Start_GRCh38, End_GRCh38 – genomic coordinates of the MPRA amplicons;Amp_name – amplicon name, including amplicon number; Maxi_counts – pre-viral Maxiprep library counts;L1_DNA_count, L2_DNA_count, L3_DNA_count, L4_DNA_count – DNA counts;L1_RNA_count, L2_RNA_count, L3_RNA_count, L4_RNA_count – RNA counts;L4_RNA_count_35 – RNA counts of a technical replicate subjected to 35 PCR cycles;Cloned_successfully – a Boolean flag indicating amplicons with at least 200 counts in the pre-viral Maxiprep library.

• Supplementary file 4. lm_308_data.csv. Amp_number – unique amplicon number; Amp_name - amplicon name, including amplicon number; Chr, Start, End – genomic coordinates of the MPRA amplicons, GRCh38 coordinates; Group – amplicon group: LD, GWAS, FBDHS, or PutEnh; L1_DNA_count, L2_DNA_count, L3_DNA_count, L4_DNA_count – DNA counts; L1_RNA_count, L2_RNA_count, L3_RNA_count, L4_RNA_count – RNA counts; DNA_count_mean, DNA_count_SD, RNA_count_mean, RNA_count_SD – mean and standard deviation of DNA and RNA counts; L1_DNA_prop, L2_DNA_prop, L3_DNA_prop, L4_DNA_prop – DNA proportions; L1_RNA_prop, L2_RNA_prop, L3_RNA_prop, L4_RNA_prop – RNA proportions; DNA_prop_mean, DNA_prop_SD, RNA_prop_mean, RNA_prop_SD – mean and standard deviations of DNA and RNA proportions; L1_Activity, L2_Activity, L3_Activity, L4_Activity – ratiometric activity; Mean_act, Mean_act_SD – mean and standard deviation of ratiometric activity; Maxi_counts, Maxi_prop – pre-viral Maxiprep counts and proportions; GC – GC content in the amplicon sequence; MeanRatio_MoM – mean ratiometric activity calculated as RNA_prop_mean / DNA_prop_mean; MeanRatio, MeanRatio_sd – mean ratiometric activity calculated as means of L1 - L4_Activities, and its standard deviation; Act_Rank – rank of the mean ratiometric activity MeanRatio_MoM; Residuals_Manual – linear model residuals; Residuals_Z_scaled_to_lm – Z-scaled linear model residuals; Pvalue – P values of the lm residuals; qval_lfdr_two_tailed – empirical tail area-based FDR calculated using lfdrtool R package; qval_lfdr_one_tailed_significant_at_0.1 – Boolean flags indicating significance at FDR < 0.1; Pvalue_two_tailed – two-tailed P values; BH_FDR_two_tailed - Benjamini-Hochberg adjusted P values, two-tailed; BH_FDR_one_tailed_significant_at_0.1 – Boolean flags indicating significance at FDR < 0.1; Pvalue_WRS_two_tailed – P values of a two-tailed Wilcoxon rank-sum (WRS) test; WRS_conf_low, WRS_conf_high – confidence intervals for the WRS test; WRS_BH_FDR_two_tailed – WRS Benjamini-Hochberg corrected P values, two-tailed; WRS_sig_one_tailed – descriptors indicating the level of WRS significance;

• Supplementary file 5. Epigenomic_intersections.csv. Amp_number – unique amplicon number; Amp_name – amplicon name, including amplicon number;Group – amplicon group: LD, GWAS, FBDHS, or PutEnh;Residuals_Z_scaled_to_lm – Z-scaled linear model residuals; Pvalue – P values of the linear model residuals; DNase_Roadmap, H3K27me3_Roadmap… – Boolean flags (columns) indicating amplicon intersections with MPRA amplicons (rows); TRUE = amplicon interval intersects with the epigenomic mark.

• Supplementary file 6. Epigenomic_predictors_of_activity.csv. Epigenomic_predictor – epigenomic mark;Epi_GLM_covariate_P_value – T-test P value of the general linear model (GLM) epigenomic covariate;P_0.05, P_0.05_Bonf.corr – Boolean flags indicating P value significance, without and with Bonferroni correction;Point_biserial_cor – Point-biserial correlations between the Residuals_Z_scaled_to_lm and epigenomic marks. Positive values indicate positive correlations with epigenomic signatures;BIC_without_epi_mark – Bayesian information criterion (BIC) values evaluating a general

linear model (GLM) without the epigenomic covariate;BIC_with_epi_mark – BIC values evaluating GLMs with each epigenomic covariate;Model_improved_by_BIC – Boolean flags indicating if a GLM prediction improved (reduced BIC) following the epigenomic covariate addition to the model.

• Supplementary file 7. SNP_All_440.csv. Amp_number – unique amplicon number;Name - amplicon name, including amplicon number;Group2 – a secondary MPRA amplicon group designator;Group – amplicon group: LD, GWAS, FBDHS, or PutEnh;chr, position – SNP genomic coordinates, GRCh38 genome; SNP_ID – SNP identifiers;reference_base – wild type allele base at the SNP genomic position, GRCh38 genome;Maxi_depth, Maxi_ref_freq – SNP read coverage and the fraction of the reference allele base at the SNP position in the pre-viral maxiprep library;L1_DNA_depth, L1_DNA_ ref_freq…L1_RNA_depth, L1_RNA_ref_freq… - – SNP read coverage and the fraction of reference allele bases at the SNP position in the RNA and DNA samples;Min_DNA_depth – a Boolean flag indicating if a SNP is covered by at least 500 reads in all 4 DNA samples;Min_RNA_depth – a Boolean flag indicating if a SNP is covered by at least 1,000 reads in all 4 RNA samples;Minor_allele_ freq_threshold – a Boolean flag indicating if a SNP has minor allele frequency over 0.05.

• Supplementary file 8. STAR408_GEO_Metadata.xlsx. Gene Expression Omnibus metadata file including sample, sequencing library, and fastq file characteristics.

• Supplementary file 9. In silico PCR.csv. UNIQID – amplicon name, including amplicon number;Left. Primer / Right.Primer – PCR primer sequences;Tm.Left / Tm.Right – melting temperature of the primers [°C];Amplicon.Length – PCR amplicon length [bp];Primer.ID - unique amplicon number;Amplicon.Range_hg19 – genomic coordinates, hg19 genome;Group - a secondary MPRA amplicon group designator;Notes – notes about the primer design;Chr – chromosome of the designed amplicon;Chr_GRCh38_pred, Amplicon_Start_GRCh38,Amplicon_End_GRCh38 – genomic coordinates of the predicted in-silico PCR products;Amplicon_length_GRCh38 – in-silico PCR predicted amplicon length;PCR_efficiency – predicted in-silico PCR efficiency;Sequence_GRCh38 – amplicon sequence;GC_content – GC content in the amplicon sequence.

• Transparent reporting form

## Data availability

All supplementary information, including links to raw and processed data, can be found at the Nord Lab Resources page (https://nordlab.faculty.ucdavis.edu/resources/). Software can be found at the Nord Lab Git Repository (https://github.com/NordNeurogenomicsLab/) and https://github.com/Nord-NeurogenomicsLab/Publications/tree/master/Lambert_eLIFE_2021,(copy archived at swh:1:rev:5f-884418d28e2baba09b91efc264b294a246f6a3). Sequencing data have been deposited in GEO under accession code GSE172058.

The following dataset was generated:

| Author(s) | Year | Dataset title | Dataset URL | Database and Identifier |
| --- | --- | --- | --- | --- |
| Lambert JT, Su-Feher L, Cichewicz K, Warren TL, Zdilar I, Wang Y, Lim KJ, Haigh J, Morse SJ, Canales CP, Stradleigh TW, Castillo E, Haghani V, Moss S, Parolini H, Quintero D, Shrestha D, Vogt D, Byrne LC, Nord AS | 2021 | Parallel functional testing identifies enhancers active in early postnatal mouse brain | https://www.ncbi.nlm.nih.gov/geo/query/acc.cgi?acc=GSE172058 | NCBI Gene Expression Omnibus, GSE172058 |

The following previously published datasets were used:

| Author(s) | Year | Dataset title | Dataset URL | Database and Identifier |
|---|---|---|---|---|
| Roadmap C | 2013 | Roadmap Consolidated Peak Dataset | https://egg2.wustl.edu/roadmap/data/byFileType/peaks/consolidated/narrowPeak/ | GEO GSM530651, GSM595913 |
| Fullard JF, Hauberg ME, Bendl J, Roussos P | 2018 | An Atlas of Chromatin Accessibility in the Adult Human Brain | https://www.ncbi.nlm.nih.gov/geo/query/acc.cgi?acc=GSE96949 | NCBI Gene Expression Omnibus, GSE96949 |
| Vierstra J, Lazar J, Sandstrom R | 2020 | Global reference mapping of human transcription factor footprints | https://www.vierstra.org/resources/dgf | N/A, resources |

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
