## [Decision Letter]

**Acceptance summary:**

This study identifies functional enhancers in vivo in the postnatal mouse brain using massively parallel reporter assays. The authors also carry out a number of validation assays to support their findings and show how the approach can be generalizable to other questions.

**Decision letter after peer review:**

Thank you for submitting your article "Parallel functional testing identifies enhancers active in early postnatal mouse brain" for consideration by *eLife*. Your article has been reviewed by 3 peer reviewers, and the evaluation has been overseen by a Reviewing Editor and Kathryn Cheah as the Senior Editor. The following individuals involved in review of your submission have agreed to reveal their identity: Stefan Barakat (Reviewer #1); Joseph D Dougherty (Reviewer #2).

Essential revisions:

Please either provide additional experimentation to address these two points or at least discuss why they should be done but you have not provided that data in this manuscript.

1) Additional negative controls.

2) Assessment of CACNA1C variant activity.

*Reviewer #1 (Recommendations for the authors):*

The authors perform an in vivo screening to identify functional enhancers in early postnatal mouse cortex by means of a massively parallel reporter assay (MPRA). To this end they designed an MPRA library containing 408 human DNA sequences with potential regulatory activity in the brain and choose an AAV vector for delivery of the reporter library. With this approach 41 amplicons with regulatory activity were identified, including the ones located in the disease-associated third intron of CACNA1C. By performing miniMPRA screening of orthologous mouse sequences, the authors confirm the reproducibility of the approach and provide evidence for cross-species conservation of regulatory elements' function. In a series of confirmation experiments the authors validate the activity of a small number of individual amplicons and, also show that the cell type specificity of a small number of investigated enhancers is preserved in this reporter assay. The described methodology might be instrumental for further in vivo studies of regulatory elements.

Strengths:

The study successfully implements a MPRA in an in vivo system and highlights considerations for the experimental design.

Weaknesses:

As also noted by the authors in the discussion, the approach does not yet allow to compare the activity of sequence variants in this assay, which will be crucial for understanding the effects of variants on regulatory function. The number of regions assessed is relatively small, and this might be a bottleneck to apply this approach to study regulatory elements at large scale.

I would like to congratulate Nord and colleagues for an interesting application of STARR-seq in vivo. I think the manuscript is definitively interesting, but sometimes I feel the statements are a bit overrated, and I would suggest that the authors tone down a bit on some of their claims. Also, there could be a few experiments added to further strengthen the quality of the paper.

My specific points, in order of appearance in the manuscript:

– If I understand correctly, the authors start off with 408 regions of interest that they start to clone, but then in line 108 they say that they could upon batch cloning, only verify the presence of 345 of these regions. Why is that? Just failed PCRs for the remaining 63 regions? Is this introducing any bias that should be explained perhaps? Perhaps also good to keep in perspective that 408 regions of interest is rather small, and finding 41 of them as active is not a lot (also given that there were several layers of pre-selection for putative regulatory regions to reach to the 408 regions); Just in general, one might think how many enhancers are required, to put on the label "massively parallel reporter assay"; a question that perhaps also remains unanswered in the manuscript is how likely the author find it that such an approach can be further up-scaled, to investigate the many millions of putative enhancers or GWAS SNPs that are there for human brain?

– In line 94-99, the authors describe the different groups from which their regions of interest were selected. Amongst this, DNaseI hypersensitivity sites were chosen. In line 170, the authors then mention that amongst the region they found active, there was a significant enrichment for loci characterized by DNaseI hypersensitivity. How much of that is explained by the selection bias of selecting those regions in the first place? Same for the putative enhancer group that was used; I assume the reason that these enhancers were putative enhancers in the first place, was their epigenome landscape. So is significant enrichment for H3K4me1/3 then really surprising? The authors should comment on that potential bias, and explain whether their statistics correct for this. Especially given the weight that the authors put on these findings in their conclusive sentence in line 184-186. Some of this information might be in Figure 2C, but is not really clear to me.

– In the section "Confirmation of in vivo P7 cortex MPRA enhancer results" the authors do a couple of validation experiments, to show that their STARR-seq active enhancers show cell type specific enhancer activity. Although important, I feel that the number of enhancers studied in here is very small, so I am not sure that the data will allow to justify broadly extrapolated conclusions from this handful tested sequences. Preferably, the authors should extend this analysis, including more enhancer loci, and perhaps also include experiments in which they test the orientation of the enhancer insertion into the STARR-seq plasmid. E.g., does it affect the activity and cell type specific expression if an enhancer is cloned in the sense or antisense orientation in the STARR-seq cassette? This would also be important to know if one would apply this assay at a larger scale (e.g., testing millions of sequences at ones, in which bulk cloning would probably cause different enhancer orientations in the library). In a way, it doesn't seem so surprising that a reporter assay that has already been validated in many studies (e.g. STARR-seq), seems to behave as a bona fide reporter construct (e.g., if the authors were to clone their validation enhancers in a lacZ reporter, they would probably see the same). So despite being a good control, it is also not such a "special" finding, so I think looking at the sense or antisense orientation might be more impact full?

– In the discussion (line 373-374), the authors say that their assay was not designed to directly compare sequence variants for activity; I don't see why? Would that not be something to include, for example for the CACNA1C locus, to test activity of variants in some of the SNPs in the enhancer regions they found? By including such an experiments, I think the author could increase the impact of their study, and this could be "the icing on the cake", which I feel in the current manuscript is still a bit lacking to some extent.

– Technical question: in line 494-495 the authors describes the STARR-seq library prep. In the original STARR-seq protocol from Stark and colleagues, and also later used by us and others, there is a two-step PCR protocol, with the first set of primers spanning an intron in the STARR-seq plasmid (thereby repressing any potential remaining plasmid DNA contamination) followed by a second round of nested PCR to amplify the amplicons themselves; both PCRs together have <30 cycles. In this paper, the authors use a single set of primers, and amplify both RNA and DNA samples 30x; why is that? Did they modify the original STARR-seq cassette in their adenoviral application, to no longer contain that intron design? What is the reason behind that? Could this be a reason for only a modest number of regions where they find an increased RNA/plasmid ratio? Please explain.

*Reviewer #2 (Recommendations for the authors):*

Summary: In this paper the authors develop a high throughput assay for in vivo enhancer activity, adapting an MPRA/STARR-seq AAV approach. They use ~900 bp candidate enhancers, selected from the human genome with 4 different strategies for selection, and successfully screen ~300 of these, including some from psychiatric disease loci. They then validate a handful in single construct injections. The major claims are (1 ) that they developed a 'self transcribing regulatory element MPRA strategy' for mouse forebrain. and (2) and validated enhancers that worked in mouse forebrain, including one from an intronic CACNA1C block that is in linkage with disease associated regions.

Major Strengths and Weaknesses: The strengths are that they were able to successfully screen a much larger number of enhancers in a single experiment than prior approaches, and that they included careful first-pass validation of some of these by microscopy in independent animals. The particular approach chosen, PCR cloning of amplicons, also allowed for longer elements to be studied than some of the oligonucleotide based libraries. I also thought the step during validation using co-injection of the test construct with GFP with the positive control in RFP was a very elegant way to control for locus of injection. However, there are some places where the next study building on this one might be I improved; drawbacks to the approach were (1) the lack of a good set of null sequences/negative controls in the MPRA that would have helped define what basal activity looks like. This lack made the analysis steps a bit more challenging, though their relative ranking of most active to least active elements is still likely accurate. (2) As admitted by the authors, the embedding of the enhancers in the UTR can risk conflating posttranscriptional effects of the elements (e.g. Poly A signals, miRNA binding sites) with enhancer effects (though this likely does not explain the activity of their strongest, validated elements). (3) There will be room for improvement in measurement of the enhancer activity by RNA sequencing in the assays, as correlation between biological replicates was not ideal (.54-.842). The noise in the assay can likely be overcome with additional replicates or additional improvements in library delivery or recovery steps. Such improvements might also allow for an increase in the number of elements assessed in parallel.

Likely impact: I think the study demonstrates feasibility for assaying hundreds of assays in parallel for activity in the brain (and similar approaches could be taken for other tissues), and provides a good foundation for future improvements to similar approaches. As it stands, it should be useful for screening more genomic loci for the most active candidate sequences. It also validates a handful of enhancers that do have function, including some in disease associated loci. These validated enhancers are potentially useful tools: understanding the cell types they express would provide some hints as to the cell types important for the disease, and having defined them also provides an opportunity to study the impact of human risk alleles on the function of these specific enhancers.

I have some concerns on the statistical approach I would like to see clarified, but I would say it is likely these are addressable. I also have some suggestions to improve flow and impact.

Concerns:

The MPRA design would have certainly benefited from a larger number of negative controls or blanks to help define which elements had enhancer activity. Something similar to https://www.pnas.org/content/110/29/11952/tab-article-info where they used scrambled sequences to understand what random genomic sequences look like, then defined enhancers based on an activity that was higher than random. That being said, the analysis they conducted does allow them to identify the sequences with the most and least relative activity. So it is probably of use even without this control if it is not feasible to add.

I could use clarification on the statistical approaches though, to be able to fully evaluate them. Use of residuals is not unheard of, but it is a bit unusual and I have a few questions I would like clarified. Specifically:

1) I am confused on the linear model as to why they authors included GC content. If, as per line ~137, no further GC bias arose after cloning, then if the model already accounts for DNA amplicon count, then the remaining effect of GC on RNA might be biological rather than technical. Perhaps high GC content correlates with more enhancer activity or RNA stability? This has certainly been seen in other studies. The authors could rerun the model without the term for GC and see how similar the results are, and see which model better predicts their existing follow up data or some of the genomic enrichment assays.

2) Overall, the authors have 2-4 different statistical approaches to picking the hits from the data they tried. Fortunately, these correspond to each other fairly well. Still, it might be helpful to use some kind of benchmarks (e.g., enrichment for overlap with the epigenomic datasets perhaps?) to determine which ought to be the primary analysis that is presented.

3) Line 539 I could also use a more clear explanation (or reference) to explain how they convert the Z-score of the residual into a p-value.

4) And, when conducting the Wilcoxon rank sum, what are they comparing each enhancer to?

5) In the end, regardless of which statistical method they lead with, they ought to make sure to also correct for multiple testing. It is not clear this was done here.

Suggestions to improve the impact:

The library is actually made of 4 sub-libraries. Are there any differences in activity between the four sub libraries? Is there anything we can learn about the relationship of LD SNPs to function, or the efficicacy of epigenetic marks at predicting function by comparing the 'hit rates' between these 4 libraries?

It is also unclear if they ended up with multiple alleles in the cloning. They cloned from a pool of DNA, so presumably different haplotypes were present. If so, were they powered to look at the allelic effects of in the functional enhancers? Or assess any in follow up?

Finally, it would bolster their second claim a bit if they did some more in depth characterization of the enhancers they validated. Are they expressed only in neurons, or neurons and glia? Are they specific to a subpopulation of neurons (at least excitatory vs. inhibitory?)? Recent comparable AAV/enhancer screens have done more in this regard, and it could be a nice deliverable of this paper if one of these enhancers happens to target a useful subpopulation. Or if it helped us better understand the regulation of some of the psychiatric disease genes. That would improve impact and it seems they have the reagents in hand to do these studies.

Suggestions on writing:

Regarding figure 1B – are the elements with sig negative residuals thought to be repressors? It might be worth discussing. Are they enriched in any particular marks?

It is good the authors discussed some limitations to cloning candidate enhancers into the 3'-UTR position, as they might have some effects on the RNA level via post-transcriptional regulation. Also, sequences containing elements that would have obvious negative consequences (e.g., poly A signals) on the assay might need to be filtered out, especially when looking at repressive sequences.

To my fresh eyes, it sometimes feels experiments are a bit out of order. For example, some of the controls showing the method should work come after the experiment rather than before it. Specifically, it is just a suggestion, but some of the experiments like those establishing that DLX works regardless of whether it is in 3'-UTR of 5' to the promoter, or perhaps the miniMPRA, might make more sense presented before the main MPRA comes up?

I'd like to see a histogram of the count number of the elements in the DNA library. I am just curious about the range between the best and least cloned elements. Presumably the least well cloned elements are also the 25 % that were filtered out? Also, perhaps some scatter plot of DNA read depth vs. coefficient of variation (with color coding of the 25 % filtered out) might make a nice supplement, and provide a benchmark for future improvements to the method.

*Reviewer #3 (Recommendations for the authors):*

The article 'Parallel functional testing identifies enhancer active in early postnatal mouse brain' uses an adeno-associated virus (AAV) based high throughput approach (massively parallel reporter assay (MPRA)) to test the regulatory capacity of candidate enhancer sequences in early postnatal mouse brain. To ascertain the reproducibility of enhancer activity across MPRA studies, the authors tested four positive enhancers, two negative sequences, and orthologous mouse sequences via miniMPRA. The results these assays suggest that consistent results can be achieved through in vivo MPRA. Finally, the authors dissect regulatory elements within the CACNA1C intron that have been previously associated with disease (schizophrenia), determining that in vivo MPRA could be an efficient tool for assessing neural disease-associated regions. This paper would be of interest to the broad readership of eLife, primarily due to the use of an in vivo AAV-based brain MPRA and characterization of disease-associated regions. However, several point need to be addressed.

– A lot more information is needed about how sequences were chosen for the MPRA. The authors tested 408 sequences from four groups. How many from each group? Was there other filters to get to 408 for each group and if so what were they? Why did they shoot for this number and not more? Are there technical reasons for this? Was there overlap between the four groups and if so what was it? How did the authors choose candidate sequences in the GWAS and SNP groups? For those (GWAS and SNP) how many overlap predictive enhancer marks?

Would also mention the problem of choosing lead SNPs in text, i.e. these may not be causative SNPs, as they are just the SNPs on the genotyping array.

Would add to Figure 1A, some table/Venn diagram or other that shows the numbers of sequences that were tested for each group to make it easier for the reader to understand what was assayed.

Would mention that the fourth group was intended as a 'potential positive control', correct me if I'm wrong here.

No negative controls were tested. This is vital in most MPRAs to compare to in order to assess activity. This needs to be mentioned and discussed in detail.

– SNPs: The sequences were cloned from 'pooled human DNA'. More info on how many individuals these are, ethnicity etc. is needed here, especially as different alleles could have different function. If the applicants have variant data from their sequenced library that would be great to add. This is even more lacking in the individual testing, in particular for the CACNA1C region, where variant might affect activity. Those definitely need to have info on the haplotype actually tested d.

– The R in the correlation between RNA replicates is poor, just a little above 0.5 in some comparisons. It also appears between viral DNA and GC content. The authors need to explain this and also provide potential causes for the reproducibility of individual assays and PCR stochasticity in the library preparation and cDNA.

– Were samples 4-35 included in the subsequent studies or not? If not, what was their performance, considering the influences of the cycles in preparation. How did the authors de-duplicate reads in line 119? Were UMIs used in the experiment and if so, more info on them is needed.

– In the miniMPRA section line 188-200 it is not clear what sequences they chose? How many? What was the criteria for selecting the sequences? A brief explanation of the sequences in the main text will help the reader understand the experimental set up better. Also, more info on stats in the main text comparing between the MPRA is needed.

– Overall only a small N of sequences was tested individually, so I would reduce the tone that the individual assays validated the MPRA and make it more that these individual ones validated.

– More detailed info on the brain expression of the individual constructs is needed.

– For the CACNA1C regions, other than reporting which haplotype was tested, it would significantly improve the manuscript if both the unassociated and associated haplotype were tested to assay whether they lead to alternate function.

---

## [Author Response]

Essential revisions:Please either provide additional experimentation to address these two points or at least discuss why they should be done but you have not provided that data in this manuscript.1) Additional negative controls.

We apologize that our manuscript and analysis failed to convey that we did actually include sequences that were presumed to be negative, which made up 59 % of the tested sequences. We updated text, analyses, and figures to clarify our library design and inclusion of these presumed negatives, which were human genome regions that did not harbor predictive signatures of potential brain enhancer activity. We now show in the figures the distribution of putative positive and negative amplicons across the four amplicon ascertainment groups that made up our library, and we show that these presumed negatives indeed had a lower rate of MPRA activity (see updates to Figure 2). We also clarify that our miniMPRA included negatives as well and that these also had lower MPRA activity compared to predicted positives (updates to Figure 1). Finally, in the original submission we also validated two amplicons that were identified as MPRA negative, showing that these sequences did not drive GFP in P7 mouse brain, in contrast to positives that did drive GFP (Figure 4 and 5).

Discussing this in more detail, there were two points that we needed to communicate more clearly, and we have now done so in the updated manuscript via revising text and Figure 2. First, the miniMPRA and the putative enhancer (“PutEnh”) group of the full MPRA included 4 negative control amplicons selected based on absence of H3K27ac in mouse brain. Second, and more relevant, our full library design did include a substantial proportion (59 %) that were putative “negative control or blanks” to enable comparison with amplicons that had some evidence for brain enhancer activity. These presumed negatives were included in the GWAS and LD amplicons selected based on their linkage to GWAS-associated SNPs. These amplicon sets contrast to our “FBDHS” putative positive amplicons, selected to be enriched with fetal brain DNAse hypersensitive signal (FBDHS). 64 % FBDHS set amplicons intersect with DHS peaks (those that didn’t had sub-significant signal enrichment), compared to 19 % and 13 % of the GWAS and LD amplicons, respectively. While not used for ascertainment, we also intersected our amplicons with a post-mortem ATAC-seq dataset from human neurons. We found that 25 % FBDHS amplicons intersect with adult neuron ATAC-seq, compared to 11 % of each GWAS and LD amplicon group. Thus, we used the LD and GWAS amplicons with no epigenomic or sequence signatures suggestive of activity as our presumed negative set.

Our MPRA library composition is explained now in Figure 2A , and the updated manuscript text now reads:

“We generated an MPRA library targeting 408 candidate human DNA sequences for testing, each ~900 bp in size, to assess functional enhancer activity in vivo in early postnatal mouse brain. […] Thus our final library included a balance of sequences that are expected to have no enhancer activity (59 %), as well as sequences with evidence for regulatory capacity in the brain (41 %).”

2) Assessment of CACNA1C variant activity.

We appreciate the general interest in testing allele-specific activity and the specific interest for the CACNA1C putative enhancers we identified. To address this issue, we include an all new analysis of SNP alleles in our experiments and we performed new luciferase experiments for one top candidate enhancer identified in the CACNA1C interval.

We note that we did perform allele-specific analysis in our MPRA, but omitted this in our original manuscript. We felt the allele-specific analysis of our experiments were not sufficiently powered to make firm conclusions and we did not want to distract from the more successful general investigation of enhancer activity in our MPRA study. However, we acknowledge that omitting the variant analysis completely would be likely to raise questions as to why it is not in the manuscript considering our design that included SNPs in the cloning. As a proposed middle ground, we describe our attempts to examine allele-specific differences in the results and discussion and show the results in a supplemental figure (Figure 2—figure supplement 4).

In this analysis, we show that our cloning strategy captured common SNP variants in our library and that variant allele frequency was highly correlated in the packaged AAV and in our MPRA DNA preps, though more variable in our RNA. We show how using 900bp amplicons in STARR-seq orientation with a tagmentation-based library preparation results in mostly reads that were not informative for the allele, and thus we had much reduced power for variant comparison. Despite this limitation, we include results comparing allele activity for the subset of six MPRA positive enhancers that harbored SNPs, finding that differences in allele frequencies were not significant. Finally, we performed new experiments for one positive amplicon that harbors disease-relevant SNPs, comparing the activity of the two alleles in luciferase assays and finding no differences in the two human cell lines tested. In our Discussion section, we added recommendations for improving our assay for assessing differences in enhancer activity of sequence variants in vivo.

We added the following to the Results section starting at line 370 of the updated manuscript to indicate this new analysis and our conclusions:

“Lack of strong effects and insufficient power for allele-specific activity analysis

Our MPRA library included a subset of amplicons harboring SNPs that were either lead SNPs from GWAS studies (121 amplicons), or SNPs in LD with lead SNPs within associated intervals at *CACNA1C* (21 amplicons) and *SCN1A* (62 amplicons). […] We note that despite strong DNA reference allele frequency correlation in the viral library DNA, limited allele coverage, unbalanced allele frequencies, and reduced correlation of RNA allele frequencies resulted in lack of power to detect moderate to small allelic differences in our experiments (Figure 2—figure supplement 4, Supplementary Table 7).”

We modified our Discussion to reflect this analysis as well, and we further discussed the limitations of our current study design and suggestions for future work (lines 601-614 of the updated manuscript):

“Although our design was not well powered to compare the activity of sequence variants (Figure 2—figure supplement 4), future studies using allele-balanced libraries, libraries with reduced complexity, or libraries with allele-specific barcodes could greatly improve the sensitivity and power of this assay to evaluate the effects of sequence variation on enhancer activity. […] For this reason, in vivo parallel functional assays will be critical in understanding how sequence variation within enhancers contributes to altered gene expression, including for the disease-associated *CACNA1C* locus.”

We have also added the following to the Methods section (lines 776-781 and 982-999):

“Allele-specific MPRA activity

Allele-specific counts were generated from deduplicated bam files using bam-readcount (https://github.com/genome/bam-readcount) with min-base-quality set to 15, and min-mapping-quality set to 20. Allelic counts were further processed using a custom R script, applying min base quality filter of 25, and min allele frequency filter of 0.01, producing a set of 440 biallelic SNPs.”

“Luciferase assay

We constructed luciferase reporter plasmids by replacing the minP promoter in pGL4.24 (Promega) with the *Hsp1a* minimal promoter using restriction enzymes HindIII and NcoI to make pGL4.24-HspMinP. […] Firefly luciferase activities were normalized to that of *Renilla* luciferase, and expression relative to the activity of the empty pGL4.24-HspMinP vector was noted.”

Reviewer #1 (Recommendations for the authors):My specific points, in order of appearance in the manuscript:– If I understand correctly, the authors start off with 408 regions of interest that they start to clone, but then in line 108 they say that they could upon batch cloning, only verify the presence of 345 of these regions. Why is that? Just failed PCRs for the remaining 63 regions? Is this introducing any bias that should be explained perhaps?

We attempted to clone the 408 sequences at once into our MPRA backbone using batch cloning. We considered amplicons as successfully cloned if they exceed a threshold of 200 counts in the pre-viral library, and 63 amplicons were excluded from further analysis due to batch cloning failure. This threshold was chosen because it separated the bimodal distribution of counts. Cloning failure for these sequences could have occurred at any of a number of steps, including PCR, Gibson assembly, bacterial transformation, or during clonal expansion in liquid culture. Most of this is likely due to stochastic success in batch cloning.

However, to address the reviewer question, we further investigated possible sources of cloning bias. We performed in silico PCR analysis using an adapted AmplifyDNA function from the DECIPHER R package. We found that two primer pairs did not have in silico products in this analysis, and as predicted, they do not have any reads aligning to their respective regions. Eight primer pairs produced more than one in silico PCR product, with lengths permitting packaging into an AAV vector. Four of these predicted nonspecific amplicons were excluded due to low RNA or DNA abundance. The remaining 4 were detected in our library and mapped to their intended loci (none of these were MPRA active amplicons). We did not find differences in GC content or predicted amplification efficiency between the successfully and unsuccessfully cloned amplicons. Although we did not observe a bias based on any obvious sequence features, we cannot rule out this possibility. However, we note that batch cloning factors do not impact downstream analysis. We provided a supplementary table with *in silico* PCR results and added this paragraph to the Methods section:

“To consider an amplicon as present in our MPRA library, we set a threshold of at least 200 read counts per amplicon in our library sample, resulting in a reduced dataset from 408 to 345 amplicons. This threshold was established based on the distribution of library counts. No GC or sequence bias was detected following in silico PCR analysis when present and missing amplicons were compared. Two primer pairs, expected to produce PCR products in our original design using the hg19 reference genome had no predicted products in our in-silico approach, as well as in our MPRA library aligned to the GRCh38 genome (Supplementary Table 9). In silico PCR analysis was performed using a custom R script and DECIPHER R package^79^.”

Perhaps also good to keep in perspective that 408 regions of interest is rather small, and finding 41 of them as active is not a lot (also given that there were several layers of pre-selection for putative regulatory regions to reach to the 408 regions); Just in general, one might think how many enhancers are required, to put on the label "massively parallel reporter assay"; a question that perhaps also remains unanswered in the manuscript is how likely the author find it that such an approach can be further up-scaled, to investigate the many millions of putative enhancers or GWAS SNPs that are there for human brain?

There are a couple points raised here, but the main ones that are not addressed in other reviewer points seems to be: what should the bar be for MPRA status regarding number of sequences assayed and a suggestion to better discuss possibility and limitations for scaling up. First, we do not have an opinion on where the bar should be for MPRA designation, but do feel that our approach is consistent with the concept (i.e. large numbers of candidates tested in parallel) and that use of another term or qualifier would just lead to more confusion. Our “hit rate” of 41/345 (~12 %) is generally in line with other MPRA enhancer studies. We note that we included a high proportion of putative negatives so “pre-selection” included both positive and negative signatures, as described in depth in other reviewer points. Nonetheless, we acknowledge that our study is on the smaller end of MPRAs, and we are open to other descriptors though our opinion is that using the term MPRA will be of greatest utility for literature searches. Second, there are very different considerations for parallel testing (i.e. MPRA) in the context of a heterogeneous mammalian tissue, made further challenging by methods for delivery that yield sparse cell uptake. We did updated our discussion to highlight these limitations.

– In line 94-99, the authors describe the different groups from which their regions of interest were selected. Amongst this, DNaseI hypersensitivity sites were chosen. In line 170, the authors then mention that amongst the region they found active, there was a significant enrichment for loci characterized by DNaseI hypersensitivity. How much of that is explained by the selection bias of selecting those regions in the first place? Same for the putative enhancer group that was used; I assume the reason that these enhancers were putative enhancers in the first place, was their epigenome landscape. So is significant enrichment for H3K4me1/3 then really surprising? The authors should comment on that potential bias, and explain whether their statistics correct for this. Especially given the weight that the authors put on these findings in their conclusive sentence in line 184-186. Some of this information might be in Figure 2C, but is not really clear to me.

We apologize that we failed to convey design considerations for sequences to include. This point is also relevant to essential revision #1, as our amplicon ascertainment groups each had a different set of criteria. To answer this specific point, only one group of our test sequences was selected based on Fetal brain DNAseI hypersensitivity (FBDHS) signal. This group (labeled FBDHS) made up 129/408 (31.6 %). We also included a small set of sequences that were screened in our miniMPRA, which included presumed positive/negatives based on presence/absence of mouse forebrain H3K27ac. The other two ascertained groups were selected based on overlap with lead SNPs (171/408, 41.9 %) or SNPs in two LD blocks (92/408, 22.5 %) from GWAS studies of schizophrenia and epilepsy. For these SNP-based amplicon sets, most do not overlap with epigenomic signatures indicating brain enhancer activity such as fetal brain DHS (81.3 % for GWAS; 87.0 % for LD) or adult neuron ATAC-seq (88.9 % for GWAS; 89.1 % for LD). This design enabled us to compare sequences that had FBDHS (and other relevant epigenomic datasets, e.g. H3K4me1, etc.) with those that did not have these signatures. Hopefully this response and our revised manuscript better explain our study design and why there is no bias in the MPRA versus epigenomic comparisons. We believe that the question of how MPRA-based activity compares to FBDHS and other enhancer signatures is fair and of great interest.

– In the section "Confirmation of in vivo P7 cortex MPRA enhancer results" the authors do a couple of validation experiments, to show that their STARR-seq active enhancers show cell type specific enhancer activity. Although important, I feel that the number of enhancers studied in here is very small, so I am not sure that the data will allow to justify broadly extrapolated conclusions from this handful tested sequences. Preferably, the authors should extend this analysis, including more enhancer loci, and perhaps also include experiments in which they test the orientation of the enhancer insertion into the STARR-seq plasmid. E.g., does it affect the activity and cell type specific expression if an enhancer is cloned in the sense or antisense orientation in the STARR-seq cassette? This would also be important to know if one would apply this assay at a larger scale (e.g., testing millions of sequences at ones, in which bulk cloning would probably cause different enhancer orientations in the library). In a way, it doesn't seem so surprising that a reporter assay that has already been validated in many studies (e.g. STARR-seq), seems to behave as a bona fide reporter construct (e.g., if the authors were to clone their validation enhancers in a lacZ reporter, they would probably see the same). So despite being a good control, it is also not such a "special" finding, so I think looking at the sense or antisense orientation might be more impact full?

We appreciate the general point here, but believe that further validation work examining more loci in different orientations is outside the scope of our study. Questions of STARR-seq performance and assay validity have been looked at by others in depth in cell lines, where it is much more tractable to do these types of experimental permutations. Indeed, there is expected to be some differences in assay response depending on sequence and orientation context. Regardless, published studies have shown the validity of STARR-seq overall and shown general concordance compared with other MPRA designs. Nonetheless, we agree that there remain questions of how MPRA results are impacted by construct design and experimental methods but argue that those questions are outside the focus and scope of our study.

We felt it most critical to address two basic items specifically relevant to in vivo activity in our validation experiments. The first, was whether cell-type specificity in vivo in brain was preserved in the STARR-seq orientation, which we did using a well characterized Dlx5/6 interneuron enhancer (now shown in Figure 1). The second was to show that our MPRA library results reproduce when sequences were tested individually for both positive and negative sequences (now shown in Figures 3 and 4). These findings are not presented as particularly special findings, just as experimental support for rigor and reproducibility. Significant effort went into these validation experiments. Each candidate sequence that is tested individually requires cloning, packaging, injection to neonatal mouse brain, tissue collection and processing, and image analysis. We feel that our study goes further towards validation experiments than many similar studies. We validated similar performance between orientations for one enhancer and we validated three positive and two negative predictions. Adding one or two more validation experiments to look at a small number of additional known or putative enhancers would require significant time and resources and does not seem like it would address this reviewer point that there are likely to be sequences that show more inter-assay differences or assay bias than the ones we tested.

We have updated the manuscript to limit the scope of our conclusions based on our validation experiments, and we have also moved our testing of 5’ vs. 3’ orientation to a new section titled “Developing and Validating AAV-MPRA strategy” where we describe our efforts to validate the appropriateness of the STARR-seq strategy to the in vivo mouse brain context before cloning the library (lines 85-178 in the updated manuscript). We also revised discussion text to frame these issues, describing the need to extend studies to characterization of different enhancer orientations and interactions with different minimal promoters.

The passage in the results referenced by Reviewer #1 in the above comment now reads:

“To validate enhancer activity in the mouse brain at P7, we next cloned individual amplicons for selected positive and negative hits from the MPRA into the same HspMinP-EGFP 3’-UTR oriented reporter and generated scAAV9 for each construct. […] These results demonstrate that our MPRA could accurately reflect enhancer activity of particular candidate sequences in vivo.”

– In the discussion (line 373-374), the authors say that their assay was not designed to directly compare sequence variants for activity; I don't see why? Would that not be something to include, for example for the CACNA1C locus, to test activity of variants in some of the SNPs in the enhancer regions they found? By including such an experiments, I think the author could increase the impact of their study, and this could be "the icing on the cake", which I feel in the current manuscript is still a bit lacking to some extent.

See our response in Essential Revision #2 for details. Briefly, we agree that allele-based enhancer activity differentiation is of high interest and is an exciting potential application of our MPRA methods. We indeed included alleles in our library which potentially could have displayed allele-specific differences in activity, and we explored this when analyzing our MPRA data. We note that our primary goal was to test for enhancer activity overall, with the hope that we would have power to compare alleles as well. We feel that our power for allelic comparison was not sufficient to highlight findings. However, we now include this analysis of allele composition and allele-based differences in enhancer activity and have added luciferase testing of one candidate amplicon comparing reference and variant alleles. These analyses and findings are now summarized in the text and in new Figure 2—figure supplement 4.

– Technical question: in line 494-495 the authors describes the STARR-seq library prep. In the original STARR-seq protocol from Stark and colleagues, and also later used by us and others, there is a two-step PCR protocol, with the first set of primers spanning an intron in the STARR-seq plasmid (thereby repressing any potential remaining plasmid DNA contamination) followed by a second round of nested PCR to amplify the amplicons themselves; both PCRs together have <30 cycles. In this paper, the authors use a single set of primers, and amplify both RNA and DNA samples 30x; why is that? Did they modify the original STARR-seq cassette in their adenoviral application, to no longer contain that intron design? What is the reason behind that? Could this be a reason for only a modest number of regions where they find an increased RNA/plasmid ratio? Please explain.

We did not include the intron in our vector design. Rather than use a nested PCR strategy to eliminate viral DNA contamination from the RNA samples, we treated these samples with DNase during the RNA purification procedure. We elected not to include the 130-230 bp chimeric intron in the vector design so that we could maximize the length of amplicons in our library (~900 bp). Eliminating the two-step PCR also enabled us to use a higher cycle count PCR program to amplify the target viral DNA and RNA sequences because the mouse brain samples included many cells that were not transduced by the library vector and therefore genomic DNA and cellular RNA represented the majority of molecules in our DNA and RNA preparations. We found we needed the higher cycle count to obtain DNA or cDNA amplicons with high enough concentration and purity for sequencing. We do not believe that viral DNA contamination of our RNA preparations explains the modest number of significant MPRA hits; as we have clarified above, the majority of our test library was expected to be negative for enhancer activity.

We have added specific details about how our vector backbone was designed and cloned in the updated manuscript:

“We replaced the CMV-EGFP expression cassette in scAAV-CMV-EGFP^71^ with a synthesized gBlocks Gene Fragment (IDT) containing the *Hsp1a* minimal promoter followed by the gene for EGFP with a cloning site inserted into the 3’-UTR. This cloning site consisted of two restriction sites flanked by two 35 bp sequences used for Gibson homology. The resulting plasmid, pscAAV-HspMinP-EGFP, housed the 3’-UTR-oriented enhancer reporter construct between the ITR sequences necessary for packaging scAAV particles (Figure 1—figure supplement 1). Unlike other STARR-seq constructs^24,72,73^, we did not include an intronic sequence in the reporter ORF, as we preferred to reduce construct length and complexity considering the novel implementation in an scAAV context and 900 bp amplicon sequence.”

Reviewer #2 (Recommendations for the authors):I have some concerns on the statistical approach I would like to see clarified, but I would say it is likely these are addressable. I also have some suggestions to improve flow and impact.Concerns:The MPRA design would have certainly benefited from a larger number of negative controls or blanks to help define which elements had enhancer activity. Something similar to https://www.pnas.org/content/110/29/11952/tab-article-info where they used scrambled sequences to understand what random genomic sequences look like, then defined enhancers based on an activity that was higher than random. That being said, the analysis they conducted does allow them to identify the sequences with the most and least relative activity. So it is probably of use even without this control if it is not feasible to add.

See our Essential Revisions (1) for this point. Briefly, we did indeed include a presumed negative set, but obviously failed in communicating this in our original submission. We added a new figure panel and updated the text to address this.

I could use clarification on the statistical approaches though, to be able to fully evaluate them. Use of residuals is not unheard of, but it is a bit unusual and I have a few questions I would like clarified. Specifically:1) I am confused on the linear model as to why they authors included GC content. If, as per line ~137, no further GC bias arose after cloning, then if the model already accounts for DNA amplicon count, then the remaining effect of GC on RNA might be biological rather than technical. Perhaps high GC content correlates with more enhancer activity or RNA stability? This has certainly been seen in other studies. The authors could rerun the model without the term for GC and see how similar the results are, and see which model better predicts their existing follow up data or some of the genomic enrichment assays.

We agree that GC content could theoretically impact enhancer activity or RNA stability, which was why we looked at it in the first place. On balance, we thought including the term had more value, as we were more concerned with bias arising from a GC impact that was not associated with enhancer activity (e.g. impact on RNA stability). That said, the inclusion or exclusion of a GC term in the regression model had very little impact on which amplicons were deemed statistically significant using model residuals.

As requested, we include a comparison of linear models constructed with and without the GC content, we found that the addition of the GC content is a significant covariate in the model (P < 1.42e-08), and inclusion increases the adjusted R^2^ from 0.7012 to 0.7369 and reduces the Bayesian information criteria (BIC) from 626.2976 to 599.1863. The P value of the GC content is 1.416e-08 (F = 34.446), whereas the P value for the log2 DNA proportions is < 2.2e-16 (F = 659.440), indicating that although statistically significant, the GC content term has minor effect on the model. We opted to leave GC in our modeling but include Figure 2–Supplementary Figure 3 panel G and H now, demonstrating a linear model with and without the GC content covariate.

“GC content had a significant impact, but low effect size, on the model as demonstrated by higher P values, 1.416e-08 vs < 2.2e-16, and lower F statistics, 34.446 vs 659.44, when log2 DNA proportions were compared with the GC content. Inclusion of the GC content was justified based on the reduction of Bayesian information criterion (BIC), from 626.2976 to 599.1863, for models without and with GC content, respectively.”

In the figure caption, we explained the justification for including the GC content in the model:

“(G) Linear model summary for estimate of background activity without the GC content covariate. (H) Linear model summary for estimate of background activity including the GC content covariate. Addition of the GC content covariate reduces the Bayesian information criteria (BIC) from 626 to 599 indicating that GC content is a valid covariate in the model.”

2) Overall, the authors have 2-4 different statistical approaches to picking the hits from the data they tried. Fortunately, these correspond to each other fairly well. Still, it might be helpful to use some kind of benchmarks (e.g., enrichment for overlap with the epigenomic datasets perhaps?) to determine which ought to be the primary analysis that is presented.

We apologize for any confusion here. Our intent was to increase statistical rigor via using multiple methods for determining MPRA activity. We note that we think the regression model residuals method is the strongest and most conservative (i.e. fewer positive calls), as this method effectively corrects for input DNA and makes no assumption of equal baseline representation in DNA and RNA libraries (which ratiometric comparison does). Nonethless, we also used the simple ratiometric method, as this method is more widespread in published MPRAs and enables inclusion of the biological replicates.

We consider the in vivo single-candidate validation of our MPRA to be the strongest validation of our approach, and we hoped that using multiple statistical approaches for MPRA activity calculation would make our analysis more transparent to readers coming from different backgrounds or with different familiarity with MPRA methods. As the reviewer notes, the two methods were overall highly concordant. We revised the section of the results to try to better explain our use of the two methods and our explanation that the resulting active amplicons were very similar.

Regarding benchmarking, the residual and ratiometric methods were similar enough and our MPRA library small enough that there wasn’t really anything learned by comparing statistical methods. That said, we did follow the advice to try benchmarking against the epigenomic signatures via including these in our regression model and seeing if the epigenomic evidence improves prediction of MPRA RNA levels. We used the full dataset of 308 amplicons that met the minimum read count thresholds. We then added epigenetic features, one at a time, in addition to the GC content covariate, and tested if the epigenetic covariate tests as significant in the model. The improvement of the model prediction was tested by assessing the reduction of the bayesian information criterion (BIC) upon the addition of an epigenomic mark.

The following fragment in the Results section addresses the predictive power of epigenomic covariates in our MPRA:

“We also examined each epigenetic mark as a co-variate of a general linear model, to see if it improved a regression model predicting cDNA levels considering all amplicons (N = 308). If cDNA levels in our MPRA are reflective of true enhancer activity, we would expect epigenomic signatures associated with enhancer elements to improve prediction of cDNA expression. Indeed, Fetal DNase hypersensitivity (P = 0.0002), H3K4me1 (P = 0.0002), adult neuron ATAC-seq (P = 0.0073), and fetal brain TF footprints (P = 0.04) were found to be significant, while signatures not relevant to enhancer activity in brain were not (Supplementary Table 5 and 6). Fetal brain DNase hypersensitivity, H3K4me1, and human neuron ATAC-seq improved the regression model (reduced BIC), and fetal brain DNase hypersensitivity and H3K4me1 signatures had the strongest predictive power of amplicon activity (0.2 and 0.178 point-biserial correlations, respectively).”

Supplementary Table 5 includes the details of this analysis.

Methods section has been updated as well:

“The predictive power of epigenomic signatures for MPRA activity was tested by adding the dichotomous Boolean variables of epigenomic intersection to a general linear model (GLM) predicting the level of RNA from DNA, and a GC content covariate, one epigenomic intersection at a time using the R stats glm() function. T test P values were extracted from model summary, and models with and without the epigenomic covariates were compared using BIC. In addition to the GLM, we calculated point-biserial correlations between the Z-scaled residuals from our original lm model, and each epigenomic signature intersection using the biserial.cor() function from the ltm R package^82^.”

3) Line 539 I could also use a more clear explanation (or reference) to explain how they convert the Z-score of the residual into a p-value.

Z-score and resulting P-values were generated using standard approaches where residual values were centered and scaled using mean and standard deviation, and then p-values were calculated using an assumed normal distribution. We have clarified our methodology in the updated manuscript by adding the underlined fragments and a citation:

“The model was applied to the complete dataset, and one-tailed p-values were calculated from the Z-scaled distribution of residuals, using the normal distribution implemented using R stats pnorm()^80^.”

4) And, when conducting the Wilcoxon rank sum, what are they comparing each enhancer to?

We compared the proportion reads in RNA libraries in each biological replicate to the proportion reads in DNA for each library using the standard rank sum test to compare these distributions. We added multiple testing correction as suggested by the reviewer. We clarified this approach in the updated manuscript. This section (lines 825-828) now reads:

“We also compared our linear model with a Wilcoxon rank sum, a non-parametric test comparing DNA vs RNA proportions, using Benjamini-Hochberg FDR < 0.05 for multiple testing correction.”

5) In the end, regardless of which statistical method they lead with, they ought to make sure to also correct for multiple testing. It is not clear this was done here.

We would like to thank Reviewer #2 for bringing up this concern. We now include multiple testing corrections, using an empirical tail area-based FDR for the residual model method and Benjamini-Hochberg FDR for the ratiometric wilcoxon tests. See results text and figures for inclusion of FDR values. We updated our methods with the following text:

“Empirical tail area-based FDR q values, estimated from Z scores, were calculated using fdrtool R package^81^, with threshold adjustment for one-tail testing.”

Suggestions to improve the impact:The library is actually made of 4 sub-libraries. Are there any differences in activity between the four sub libraries?

As expected, the four amplicon groups in our MPRA library (FBDHS, GWAS, LD, PutEnh) differ in average amplicon activity, with the highest values reported for the PutEnh group, followed by the FBDHS group. In contrast, the amplicons ascertained just on presence of relevant SNP had lower activity. The effect of the amplicon group on the activity is statistically significant (One-way ANOVA, P = 0.006), and persists after excluding the PutEnh group (One-way ANOVA, P = 0.011).

We added the following sentence to emphasize the effect of amplicon group:

“Amplicon group was confirmed to be a significant predictor of mean amplicon activity (One-way ANOVA, P = 0.006), with the highest activity reported for the PutEnh and FBDHS groups.”

Is there anything we can learn about the relationship of LD SNPs to function, or the efficicacy of epigenetic marks at predicting function by comparing the 'hit rates' between these 4 libraries?

We respectfully direct Reviewer #2 to our response to Essential Revision #1 and #2, and Figure 2 Supplement 4 explaining how SNPs and allelic analysis were handled. See also above for our analysis on the relationship between epigenetic marks and MPRA activity.

It is also unclear if they ended up with multiple alleles in the cloning. They cloned from a pool of DNA, so presumably different haplotypes were present. If so, were they powered to look at the allelic effects of in the functional enhancers? Or assess any in follow up?

Please see our response to Essential Revisions #2. Briefly, in the updated manuscript we have added this new allelic analysis, discussed the limitations of our current study for determining allele-based enhancer activity differences, and presented suggestions for future library designs that may address these limitations (Figure 2—figure supplement 4, lines 370-386, and 601-614 of the updated manuscript). Note that we agree that this is an important future direction for this work, and we regret that our current study design was not well powered to address it.

Finally, it would bolster their second claim a bit if they did some more in depth characterization of the enhancers they validated. Are they expressed only in neurons, or neurons and glia? Are they specific to a subpopulation of neurons (at least excitatory vs. inhibitory?)? Recent comparable AAV/enhancer screens have done more in this regard, and it could be a nice deliverable of this paper if one of these enhancers happens to target a useful subpopulation. Or if it helped us better understand the regulation of some of the psychiatric disease genes. That would improve impact and it seems they have the reagents in hand to do these studies.

We agree that cell type specific characterization of enhancer activity is of significant interest, particularly to find new drivers or resolve disease-variant specificity. However, this work requires substantial experiments and further improvements in MPRA deployment in vivo in brain. As such, while of great interest, these types of studies are outside the scope of this manuscript, though we are working on these questions both at the library and single enhancer level. We note that there have been some beautiful studies recently published in these areas that we reference, and these publications represent years of work and still face issues with library complexity, relationship between epigenomic prediction to enhancer activity, and cell-type and brain region specificity.

Suggestions on writing:Regarding figure 1B – are the elements with sig negative residuals thought to be repressors? It might be worth discussing. Are they enriched in any particular marks?

It is possible that these elements are repressors, but we hesitate to draw that conclusion because of our particular study design, which inserts a relatively long sequence into the 3’-UTR and could conceivably lead to higher rates of mRNA degradation. We have expanded on this in our Discussion, which now reads:

“We detected a number of amplicons that had reduced RNA compared to expected background levels, which may represent sequences with silencer or repressor activity such as those that have been reported in recent MPRA screens^68^. However, because of both the insertion of these amplicons into the 3’-UTR and the length of the amplicons, the modified 3’-UTR may have caused the transcripts to be subject to RNA degradation. Thus, we are hesitant to draw conclusions about these amplicons with lower than expected activity without further characterization. Nevertheless, while artifacts from inserting sequences into the 3’-UTR may impact the STARR-seq design and our results, we show that orthologous mouse sequences exhibited correlated presence and absence of activity and similarly that amplicons with absence or presence of MPRA activity across replicates consistently reproduced in independent single deliveries to the brain.”

It is good the authors discussed some limitations to cloning candidate enhancers into the 3'-UTR position, as they might have some effects on the RNA level via post-transcriptional regulation. Also, sequences containing elements that would have obvious negative consequences (e.g., poly A signals) on the assay might need to be filtered out, especially when looking at repressive sequences.

We followed a few lines of analysis of sequence motifs in our amplicons, but we feel that we are vastly underpowered to look at motif analysis within the 900 bp amplicons considering the small number of high (e.g. presumed enhancers) and low (e.g. potential repressors) activity amplicons we found, and so do not report in the manuscript. We describe here to address the reviewer question.

First, we divided our data set into low, medium, and highly active amplicons and performed motif enrichment analysis using HOMER. However, no motifs were singularly associated with any activity group or were present across most amplicons in the set.

We also looked for the presence of the canonical polyA signal AATAAA in our amplicon sequences. 167/308 of our amplicons include this sequence, including 27/41 of the significantly active amplicons. We did not find a significant association between the presence of a polyA signal and the significance of amplicon activity (See distribution of PolyA-containing amplicons in Author response image 1, p-value = 0.1302, Fisher’s exact test, two-sided), and it was not a significant covariate when included in the linear model either together with the GC content (p = 0.76475), or without the GC content covariate (p = 0.0608).

**Author response image 1. sa2fig1:** 

To my fresh eyes, it sometimes feels experiments are a bit out of order. For example, some of the controls showing the method should work come after the experiment rather than before it. Specifically, it is just a suggestion, but some of the experiments like those establishing that DLX works regardless of whether it is in 3'-UTR of 5' to the promoter, or perhaps the miniMPRA, might make more sense presented before the main MPRA comes up?

We wish to thank Reviewer #2 for this suggestion. We had struggled some with how to organize our manuscript, and so opted to reorganize following the reviewer’s suggestion. We have rewritten the manuscript, presenting the mouse miniMPRA and the *Dlx* 3’ and 5’ orientation validation experiments as Figure 1 before moving to the full MPRA for the rest of the manuscript. In addition to more logically flowing from proof-of-principle to full experiment, this provided us more space to discuss our vector and library design. The reorganization resulted in a change in Figure numbering, and many figure panels have been moved or altered to fit this narrative flow. The first section of our Results now reads (lines 85-178):

“Developing and Validating AAV-MPRA strategy

We used a modified STARR-seq^24^ MPRA orientation, in which the candidates of interest were cloned in the 3’ untranslated region (3’-UTR) of a reporter gene (here EGFP) driven by the *Hsp1a* minimal promoter (HspMinP; formerly referred to as Hsp68 minP^26^). […] Based on this proof-of-principle experiment, we moved forward with scaling up our scAAV STARR-seq strategy to testing hundreds of sequences in mouse brain.”

I'd like to see a histogram of the count number of the elements in the DNA library. I am just curious about the range between the best and least cloned elements. Presumably the least well cloned elements are also the 25 % that were filtered out? Also, perhaps some scatter plot of DNA read depth vs. coefficient of variation (with color coding of the 25 % filtered out) might make a nice supplement, and provide a benchmark for future improvements to the method.

As the reviewer expects, being poorly represented in the pre-viral library (indicating a failure in cloning) corresponded with poor representation in the DNA read counts, justifying the exclusion of these amplicons from further analysis. Please see the . (Author response image 2).

Reviewer #3 (Recommendations for the authors):The article 'Parallel functional testing identifies enhancer active in early postnatal mouse brain' uses an adeno-associated virus (AAV) based high throughput approach (massively parallel reporter assay (MPRA)) to test the regulatory capacity of candidate enhancer sequences in early postnatal mouse brain. To ascertain the reproducibility of enhancer activity across MPRA studies, the authors tested four positive enhancers, two negative sequences, and orthologous mouse sequences via miniMPRA. The results these assays suggest that consistent results can be achieved through in vivo MPRA. Finally, the authors dissect regulatory elements within the CACNA1C intron that have been previously associated with disease (schizophrenia), determining that in vivo MPRA could be an efficient tool for assessing neural disease-associated regions. This paper would be of interest to the broad readership of eLife, primarily due to the use of an in vivo AAV-based brain MPRA and characterization of disease-associated regions. However, several point need to be addressed.Major comments:– A lot more information is needed about how sequences were chosen for the MPRA. The authors tested 408 sequences from four groups. How many from each group? Was there other filters to get to 408 for each group and if so what were they? Why did they shoot for this number and not more? Are there technical reasons for this? Was there overlap between the four groups and if so what was it? How did the authors choose candidate sequences in the GWAS and SNP groups? For those (GWAS and SNP) how many overlap predictive enhancer marks?Would also mention the problem of choosing lead SNPs in text, i.e. these may not be causative SNPs, as they are just the SNPs on the genotyping array.Would add to Figure 1A, some table/Venn diagram or other that shows the numbers of sequences that were tested for each group to make it easier for the reader to understand what was assayed.Would mention that the fourth group was intended as a 'potential positive control', correct me if I'm wrong here.No negative controls were tested. This is vital in most MPRAs to compare to in order to assess activity. This needs to be mentioned and discussed in detail.

We did not adequately explain our library design in our original manuscript, as is demonstrated by all three reviewers asking for clarification about the different groups of amplicons and our negative controls. We have addressed these concerns in our response to Essential Revision #1. Briefly, we have updated the manuscript with more detail on how amplicons were chosen and that many amplicons in the GWAS and LD groups were expected to be negative for enhancer activity because disease-associated SNPs may not be causal. We thank Reviewer #3 for the suggestion to add a figure panel showing this, and we have included a new figure panel (Figure 2A) which shows the relative proportions of the amplicon groups and the proportion of each group which overlaps signatures of open chromatin suggestive of potential enhancer activity in the brain. We have also clarified that the “putative enhancer” group (PutEnh) contains both positive and negative controls.

– SNPs: The sequences were cloned from 'pooled human DNA'. More info on how many individuals these are, ethnicity etc. is needed here, especially as different alleles could have different function. If the applicants have variant data from their sequenced library that would be great to add. This is even more lacking in the individual testing, in particular for the CACNA1C region, where variant might affect activity. Those definitely need to have info on the haplotype actually tested d.

We have clarified the specific pools of DNA we used in the updated manuscript. For LD amplicons, we have further clarified the alleles of each GWAS SNP that we tested in our single-candidate validation experiments. This section of our methods now reads:

“Clones were confirmed by PCR and Sanger sequencing. Amplicon #2 included the following alleles: A at rs4765904, A at rs1108221, and C at rs1108222. Amplicon #3 included the following alleles: G at rs1108075 and G at rs11062166. Amplicon #6 included the following alleles: C at rs2159100 and T at rs12315711.”

– The R in the correlation between RNA replicates is poor, just a little above 0.5 in some comparisons. It also appears between viral DNA and GC content. The authors need to explain this and also provide potential causes for the reproducibility of individual assays and PCR stochasticity in the library preparation and cDNA.

We have adjusted the text in our Results section to highlight the replicate-specific dropout of amplicons that limits the correlations between cDNA samples, and we have added more information on how GC content influences our model-based activity signal. We have also further addressed reproducibility of our assay in the Discussion section (lines 526-542 in the updated manuscript).

The passage in the Discussion now reads:

“Based on MPRA and validation experiments, the main sources of variation across MPRA replicates were AAV transduction rate and injection site (Figure 1—figure supplement 4, Figure 3—figure supplement 2, Figure 4—figure supplement 3), highlighting the need for delivery controls to ensure reproducibility when applying MPRAs in vivo via viral delivery. […] While not an obvious issue here, effect of PCR-based clonal amplification bias in other MPRAs has been shown to be reduced with the addition of barcodes and unique molecular identifiers to the reporter construct^60^.”

– Were samples 4-35 included in the subsequent studies or not? If not, what was their performance, considering the influences of the cycles in preparation.

Sample 4-35 was not included in the calculation of mean RNA proportions, which is stated in the methods section:

“For downstream analysis, mean amplicon proportions for genomic DNA (“DNA”) and cDNA (“RNA”) were calculated across all samples excluding the technical replicate Sample 4-35.”

And

“Mean ratiometric activity and standard deviation was then calculated for each amplicon across all biological replicates, excluding the technical replicate Sample 4-35.”

Sample 4-35 demonstrated the highest correlation with its technical replicate subjected to 30 PCR cycles (Figure 2 —figure supplement 1).

How did the authors de-duplicate reads in line 119?

The process of read deduplication of reads is explained in the Methods section:

“Aligned SAM files were processed using Picard tools (v.2.23.3)^53^: aligned reads were sorted by coordinates and converted to BAM files using *SortSam*, optical duplicates were flagged by *MarkDuplicates*, and BAM file indexing was conducted using *BuildBamIndex*. The number of de-duplicated reads within in silico PCR amplicon coordinates was counted using *samtools view -F 0x400 -c* (v1.7)^78^ and reported as the raw amplicon count.”

The *-F 0x400* in *samtools view -F 0x400 -c* instructs samtools view counts to ignore any reads marked as duplicates by Picard tools MarkDuplicates.

Were UMIs used in the experiment and if so, more info on them is needed.

Our experimental strategy was not designed for UMI inclusion, as we sequenced the variable amplicon using a tagmentation approach.

– In the miniMPRA section line 188-200 it is not clear what sequences they chose? How many? What was the criteria for selecting the sequences? A brief explanation of the sequences in the main text will help the reader understand the experimental set up better. Also, more info on stats in the main text comparing between the MPRA is needed.

We performed the miniMPRA as a proof-of-concept in advance of our full MPRA experiment. When we initially wrote the manuscript, we wanted to focus on the full MPRA. This led us to neglect to include important details about the miniMPRA design and rationale. We have restructured the updated manuscript, which now fully describes this experiment (see lines 162-179 in the updated manuscript). For more details on this restructuring, see our response to Reviewer #2 above.

– Overall only a small N of sequences was tested individually, so I would reduce the tone that the individual assays validated the MPRA and make it more that these individual ones validated.

We have adjusted our language in the Results and Discussion sections to limit the scope of our claims.

The Discussion section now contains the following text:

“These results suggest that the AAV MPRA implementation was effective at screening putative regulatory sequences for in vivo activity in the brain, with reporter expression concordant between MPRA results and single candidate tests of EGFP expression in P7 mouse forebrain for five individually validated amplicons, and that these results could be extended to study activity in later development.”

– More detailed info on the brain expression of the individual constructs is needed.

We have added some more detailed observations on the brain expression of the individual reporter constructs. This statement has been added to the Results section (lines 447-450):

“EGFP expression driven by this amplicon was seen wherever the brain had been exposed to the virus (based on mRuby3 expression), with a trend toward greater density of EGFP^+^ cells observed in the lower-middle layers of the cortex (lower half of Layer IV and Layer V).”

– For the CACNA1C regions, other than reporting which haplotype was tested, it would significantly improve the manuscript if both the unassociated and associated haplotype were tested to assay whether they lead to alternate function.

We agree with Reviewer #3 that this is an exciting application of the MPRA method. See our detailed response in Essential Revision #2. Briefly, when we initially designed our study, we hoped that cloning from a pooled sample of DNA representing human population variation would enable us to analyze allelic differences in amplicon activity. However, our data was not well powered for this application, largely because the number of sequencing reads that were allele-informative were a small subset of the data and the minor allele frequencies for SNPs in our library were quite low. That said, we did do this analysis and describe our results. We also performed allele-specific luciferase for one of the CACNA1C amplicons, finding no difference between the two alleles. We have also included a new supplementary figure (Figure 2—figure supplement 4) which details our analysis of allele-specific activity of amplicons in our MPRA.

We have expanded our discussion of these limitations, along with suggestions for future MPRA experiments specifically designed for the study of alleles, in the updated manuscript.